# The glucuronyltransferase B4GAT1 is required for initiation of LARGE-mediated α-dystroglycan functional glycosylation

**Tobias Willer**[1,2,3,4], **Kei-ichiro Inamori**[1,2,3,4,5], **David Venzke**[1,2,3,4], **Corinne Harvey**[1,2,3,4], **Greg Morgensen**[1,2,3,4], **Yuji Hara**[1,2,3,4,6], **Daniel Beltrán Valero de Bernabé**[1,2,3,4], **Liping Yu**[7], **Kevin M Wright**[8], **Kevin P Campbell**[1,2,3,4]*

[1]Department of Molecular Physiology and Biophysics, University of Iowa, Carver College of Medicine, Iowa City, United States; [2]Department of Neurology, University of Iowa, Carver College of Medicine, Iowa City, United States; [3]Department of Internal Medicine, University of Iowa, Carver College of Medicine, Iowa City, United States; [4]Howard Hughes Medical Institute, University of Iowa, Carver College of Medicine, Iowa City, United States; [5]Division of Glycopathology, Institute of Molecular Biomembrane and Glycobiology, Tohoku Pharmaceutical University, Komatsushima, Japan; [6]Department of Synthetic Chemistry and Biological Chemistry, Graduate School of Engineering, Kyoto University, Kyoto, Japan; [7]Medical Nuclear Magnetic Resonance Facility, University of Iowa, Carver College of Medicine, Iowa City, United States; [8]Vollum Institute, Oregon Health and Science University, Portland, United States

*For correspondence: kevin-campbell@uiowa.edu

**Competing interests:** The authors declare that no competing interests exist.

**Reviewing editor**: Suzanne R Pfeffer, Stanford University, United States

**Abstract** Dystroglycan is a cell membrane receptor that organizes the basement membrane by binding ligands in the extracellular matrix. Proper glycosylation of the α-dystroglycan (α-DG) subunit is essential for these activities, and lack thereof results in neuromuscular disease. Currently, neither the glycan synthesis pathway nor the roles of many known or putative glycosyltransferases that are essential for this process are well understood. Here we show that FKRP, FKTN, TMEM5 and B4GAT1 (formerly known as B3GNT1) localize to the Golgi and contribute to the O-mannosyl post-phosphorylation modification of α-DG. Moreover, we assigned B4GAT1 a function as a xylose β1,4-glucuronyltransferase. Nuclear magnetic resonance studies confirmed that a glucuronic acid β1,4-xylose disaccharide synthesized by B4GAT1 acts as an acceptor primer that can be elongated by LARGE with the ligand-binding heteropolysaccharide. Our findings greatly broaden the understanding of α-DG glycosylation and provide mechanistic insight into why mutations in B4GAT1 disrupt dystroglycan function and cause disease.

## Introduction

Dystroglycan (DG) is a highly glycosylated basement membrane receptor involved in a variety of physiological processes including maintenance of the skeletal muscle-cell membrane integrity and establishment of the structure and function of the central nervous system (*Barresi and Campbell, 2006*). DG is composed of a cell-surface α-subunit and a transmembrane β-subunit. α-DG acts as a receptor for laminin-G domain-containing extracellular matrix (ECM) proteins such as laminin, agrin, perlecan and neurexin (*Barresi and Campbell, 2006*). In addition, it serves as a cellular receptor and entry site for most Old World arenaviruses, including the highly pathogenic Lassa virus (LASV) and Clade C New Word arenaviruses (*Cao et al., 1998*). LASV is the causative agent of severe hemorrhagic fever in humans, a disease that has a mortality rate of ~15% resulting in several thousand deaths each year.

**eLife digest** Dystroglycan is a protein that is critical for the proper function of many tissues, especially muscles and brain. Dystroglycan helps to connect the structural network inside the cell with the matrix outside of the cell. The extracellular matrix fills the space between the cells to serve as a scaffold and hold cells together within a tissue. It is well established that the interaction of cells with their extracellular environments is important for structuring tissues, as well as for helping cells to specialize and migrate. These interactions also play a role in the progression of cancer.

As is the case for many proteins, dystroglycan must be modified with particular sugar molecules in order to work correctly. Enzymes called glycosyltransferases are responsible for sequentially assembling a complex array of sugar molecules on dystroglycan. This modification is essential for making dystroglycan 'sticky', so it can bind to the components of the extracellular matrix. If sugar molecules are added incorrectly, dystroglycan loses its ability to bind to these components. This causes congenital muscular dystrophies, a group of diseases that are characterized by a progressive loss of muscle function.

Willer et al. use a wide range of experimental techniques to investigate the types of sugar molecules added to dystroglycan, the overall structure of the resulting 'sticky' complex and the mechanism whereby it is built. This reveals that a glycosyltransferase known as B3GNT1 is one of the enzymes responsible for adding a sugar molecule to the complex. This enzyme was first described in the literature over a decade ago, and the name B3GNT1 was assigned, according to a code, to reflect the sugar molecule it was thought to transfer to proteins. However, Willer et al. (and independently, Praissman et al.) find that this enzyme actually attaches a different sugar modification to dystroglycan, and so should therefore be called B4GAT1 instead.

Willer et al. find that the sugar molecule added by the B4GAT1 enzyme acts as a platform for the assembly of a much larger sugar polymer that cells use to anchor themselves within a tissue. Some viruses–including Lassa virus, which causes severe fever and bleeding–also use the 'sticky' sugar modification of dystroglycan to bind to and invade cells, causing disease in humans. Understanding the structure of this complex, and how these sugar modifications are added to dystroglycan, could therefore help to develop treatments for a wide range of diseases like progressive muscle weakening and viral infections.

α-DG effectiveness as a receptor is dependent on complex post-translational modifications. Besides numerous modifications with *N*-glycans and mucin-type *O*-glycans, a highly complicated series of additions to a phosphorylated *O*-mannosyl glycan moiety in the N-terminal region of the mucin domain are essential for ligand binding (*Kanagawa et al., 2004*; *Hara et al., 2011*). Defects in the proper post-translational processing of α-DG result in loss of receptor function, and in a broad spectrum of congenital muscular dystrophies (CMDs) that are accompanied by a variety of brain and eye malformations. Collectively, these dystrophies are classified as dystroglycanopathies (*Barresi and Campbell, 2006*). To date, over 17 genes have been reported to be directly or indirectly involved in this 'functional glycosylation' of α-DG, and have been linked to human disease when mutated (*Mercuri and Muntoni, 2012*; *Bonnemann et al., 2014*).

Recent gene discovery efforts revealed several novel dystroglycanopathy genes with unknown function (*Vuillaumier-Barrot et al., 2012*; *Buysse et al., 2013*; *Jae et al., 2013*). In our previous work we were able to assign functions to the POMGNT2 (Protein O-linked mannose N-acetylglucosaminyltransferase 2) (GTDC2), B3GALNT2 and POMK (Protein O-mannose kinase) (SGK196) gene products, which contribute to the synthesis of a phosphorylated, O-mannosyl-linked trisaccharide on α-DG (*Yoshida-Moriguchi et al., 2013*) (*Figure 1—figure supplement 1*). This so-called Core M3 structure (GalNAc-β3-GlcNAc-β4-Man-α Ser/Thr) is synthesized in the endoplasmic reticulum (ER), and it is thought to be a platform for further functional modification of α-DG as it passes through the secretory pathway. The glycosyltransferase LARGE was shown to synthesize and transfer repeating units of [–3-xylose–α1,3-glucuronic acid-β1–] to α-DG (*Inamori et al., 2012*). This heteropolymer is postulated to be the terminal glycan moiety anchored by the Core M3 structure. It resembles the ligand-binding glycan and its length correlates with the affinity of α-DG for its ligands (*Goddeeris et al., 2013*). However, how the laminin-binding glycan synthesized by LARGE is attached to the Core M3 structure,

and which glycans or other molecules contribute to and form part of this linker structure, remains unknown.

We set out to elucidate the structure and monosaccharide composition of the α-DG post-phosphoryl modification, applying a strategic and multifaceted experimental approach starting from the terminal end synthesized by LARGE. We used glycosylation-deficient cells, in vitro enzyme assays, deglycosylation strategies, and NMR (Nuclear Magnetic Resonance)-based structure analysis as experimental tools.

This strategy revealed a β1,4 glucuronyltransferase activity for B4GAT1. We present experimental evidence that this enzyme B4GAT1, which was previously described in the literature as B3GNT1 (*Sasaki et al., 1997*) in fact encodes for a β1,4 glucuronyltransferase and not a β1,3 N-acetylglucosaminyltransferase as previously thought. This activity contributes to production of the post-phosphoryl glycan linker by transferring a glucuronic acid (GlcA) residue onto a xylose (Xyl) acceptor. It thereby forms a glucuronyl-β1,4-xylosyl disaccharide, the direct acceptor required by the glycosyltransferase LARGE to initiate formation of the terminal heteropolysaccharide that is involved in ligand binding. B4GAT1 enzymatic activity, of both a recombinant form and the endogenous protein in mouse embryonic fibroblasts (MEFs), was further characterized using a newly developed HPLC (High-Performance Liquid Chromatography)-based assay for B4GAT1 activity.

Our findings contribute to the current understanding of α-DG posttranslational processing, providing mechanistic insights regarding the pathomechanism underlying α-DG glycosylation-deficient CMDs and revealing new therapeutic avenues for blocking entry of pathogenic LASV viruses.

## Results

### α-DG O-mannosyl post-phosphoryl modification occurs in the Golgi

Our previous work showed that the ER-resident enzymes POMGNT2 (GTDC2), B3GALNT2 and POMK (SGK196) contribute to synthesis of the phosphorylated Core M3 trisaccharide on α-DG, a moiety that is required as platform for further modification with the LARGE mediated laminin-binding glycan (*Yoshida-Moriguchi et al., 2013*). However, a number of additional genes, namely *FKTN* (Fukutin) (*Kobayashi et al., 1998*; *de Bernabe et al., 2003*), *FKRP* (Fukutin-related protein) (*Brockington et al., 2001*; *Beltran-Valero de Bernabe et al., 2004*) *TMEM5* (*Vuillaumier-Barrot et al., 2012*) and *B4GAT1 (B3GNT1)* (*Wright et al., 2012*; *Buysse et al., 2013*; *Shaheen et al., 2013*) are known to be crucial for proper α-DG glycosylation, yet how they contribute has not yet been determined (*Figure 1—figure supplement 1*). To investigate if these unassigned genes are involved in the pre- or post-phosphorylation process of Core M3, we expressed Fc-tagged recombinant α-DG (DGFc340) in [$^{32}$P] orthophosphate-labeled control and glycosylation-deficient cells. DGFc340 is a secreted α-DG deletion construct that contains only the minimal region of the α-DG mucin-like domain (aa 316–340), that is required for its functional glycosylation followed by a C-terminal fusion tag encoding the heavy-chain constant (Fc) moiety of human IgG1 (to enable purification of the secreted recombinant protein) (*Hara et al., 2011*). Although only a small subpopulation of the expressed DGFc protein enters the pathway for functional maturation it was demonstrated that this truncated α-DG fusion protein is a valuable tool to study α-DG functional glycosylation (*Hara et al., 2011*).

The goal was to test if DGFc340 can be [$^{32}$P] phosphorylated in fibroblasts with defects in various dystroglycanopathy genes (*Table 1*). In our experiment, fibroblasts with defects in *FKTN*, *FKRP*, *TMEM5*, *B4GAT1 (B3GNT1)* and *LARGE*, but not in the phosphate kinase *POMK*, were able to produce radioactively labeled DGFc340 (*Figure 1A*), indicating that FKTN, FKRP, TMEM5, B4GAT1 and LARGE are involved downstream of POMK in the Core M3 post-phosphorylation process.

Immunofluorescence examination of HEK293T (Human Embryonic Kidney) cells stably transfected with Myc-tagged constructs of this set of proteins, revealed that they co-localize with the Golgi-resident marker protein Giantin (*Linstedt and Hauri, 1993*) (*Figure 1B*). Previously, Golgi localization was also demonstrated for FKRP (*Esapa et al., 2002*), FKTN (*Esapa et al., 2002*; *Xiong et al., 2006*), B4GAT1 (B3GNT1) (*Buysse et al., 2013*) and LARGE (*Brockington et al., 2005*). These results indicate that most if not all of the α-DG O-mannosyl post-phosphoryl processing is carried out by Golgi-resident enzymes.

The laminin-binding glycan repeat generated by LARGE is hypothesized to be the terminal glycan structure of the α-DG O-mannosyl post-phosphoryl modification (*Figure 1—figure supplement 1*). This would suggest that FKTN, FKRP, TMEM5 and B4GAT1 contribute to a post-phosphoryl linker structure, that can serve as an acceptor for the modification with LARGE. Previous work by Kuga et al., (*Kuga et al., 2012*) also had indicated that FKTN and FKRP are part of the α-DG O-mannosyl

Table 1. Summary of features of control and glycosylation-deficient cell lines

| Mutant gene | Clinical phenotype | Cell type | Nucleotide variant | Amino acid | reference |
|---|---|---|---|---|---|
| Control (human) | none | Human skin fibroblast | | | CRL-2127 (ATCC) |
| POMK | WWS/MEB | Human skin fibroblast | 14bp homozygous deletion (c.720_733delGCTGGTGAGTGCG)], homozygous | p.Leu241Profs*26 | (Yoshida-Moriguchi et al., 2013) |
| FKTN | WWS | Human skin fibroblast | c.385delA | p.I129fsX1 | GM16192 |
| | | | c.1176C > A, heterozygous | p.Y392X | (Coriell Cell Repository) |
| FKRP | WWS | Human skin fibroblast | c.1A > G, homozygous | p.M1V | (Van Reeuwijk et al., 2010) |
| TMEM5 | WWS | Human skin fibroblast | c.1101 G > A, homozygous | p.G333R | unpublished |
| B4gat1 (B3gnt1) | CMD | MEF | c.464T > C, compound het with LacZ null allele, B4gat1^LacZ/M155T | p.M155T | (Wright et al., 2012) |
| Large^myd | CMD | MEF | deletion of exons 5–7, homozygous | | (Grewal et al., 2001) |
| Control (mouse) | none | MEF | | | |

post-phosphoryl modification pathway. To test our hypothesis, we infected a panel of glycosylation-deficient cells with a LARGE expressing adenovirus construct and analyzed the glycosylation status of α-DG and the degree of hyperglycosylation by On-Cell immunoblotting with monoclonal antibody IIH6, which recognizes the α-DG laminin-binding glycan transferred by LARGE (Inamori et al., 2012; Goddeeris et al., 2013). As expected, overexpression of LARGE did not produce the IIH6-positive glycan or significantly bypass the glycosylation defect in either FKTN-, FKRP-, TMEM5- or B4GAT1-deficient cells (Figure 1C), supporting the notion that these encoded proteins work prior to LARGE in the O-mannosyl post-phosphoryl modification process.

In summary, our data suggest that Golgi-localized putative glycosyltransferases FKTN, FKRP, TMEM5 and B4GAT1 are essential for the synthesis of a linker structure that is connecting the α-DG O-mannosyl phosphate platform with the terminal laminin binding glycan added by LARGE.

## Glucuronic acid serves as an acceptor sugar for LARGE polymer initiation with Xylose

To further elucidate the structure of the α-DG post-phosphoryl modification, we examined how synthesis of the terminal LARGE glycan was initiated. Although LARGE is known to be a dual glycosyltransferase that synthesizes repeating units of [–3-xylose–α1,3-glucuronic acid-β1–] on α-DG, the identities of both the initiating sugar and the acceptor sugar for this laminin-binding polymer remained unknown. To determine which sugar is initially transferred by LARGE, we developed an in vitro glycosylation assay using a recombinant soluble (transmembrane domain deleted) form of LARGE (LARGEdTM) and the acceptor protein DGFc340. DGFc340 isolated from the culture medium of Large^myd (Large-deficient) MEFs lacks the LARGE modification on phosphorylated O-mannosyl glycans, and is hypothesized to terminate in a glycan acceptor structure that can be recognized by LARGE (Inamori et al., 2012) (Figure 2A). To determine which of the sugars in the polymer is initially transferred by LARGE we incubated DGFc340 isolated from the Large^myd MEF culture medium with LARGEdTM and [14C]-labeled UDP-xylose (Xyl) and/or UDP-glucuronic acid (GlcA) radionucleotide sugar donors. The glycosyl-transfer reaction was measured as the transfer of radioactivity onto the DGFc340 acceptor glycoprotein. As negative control we used a DGFc340 mutant construct (T317A/T319A), which lacks the O-mannosylation sites that are the crucial acceptor platform for subsequent synthesis of the laminin-binding glycan (Hara et al., 2011). When the radionucleotide sugars were tested individually in the LARGEdTM in vitro assay, the addition of [14C] UDP-Xyl, but not that of [14C] UDP-GlcA radionucleotides, resulted in radioactive labeling of the DGFc340 acceptor

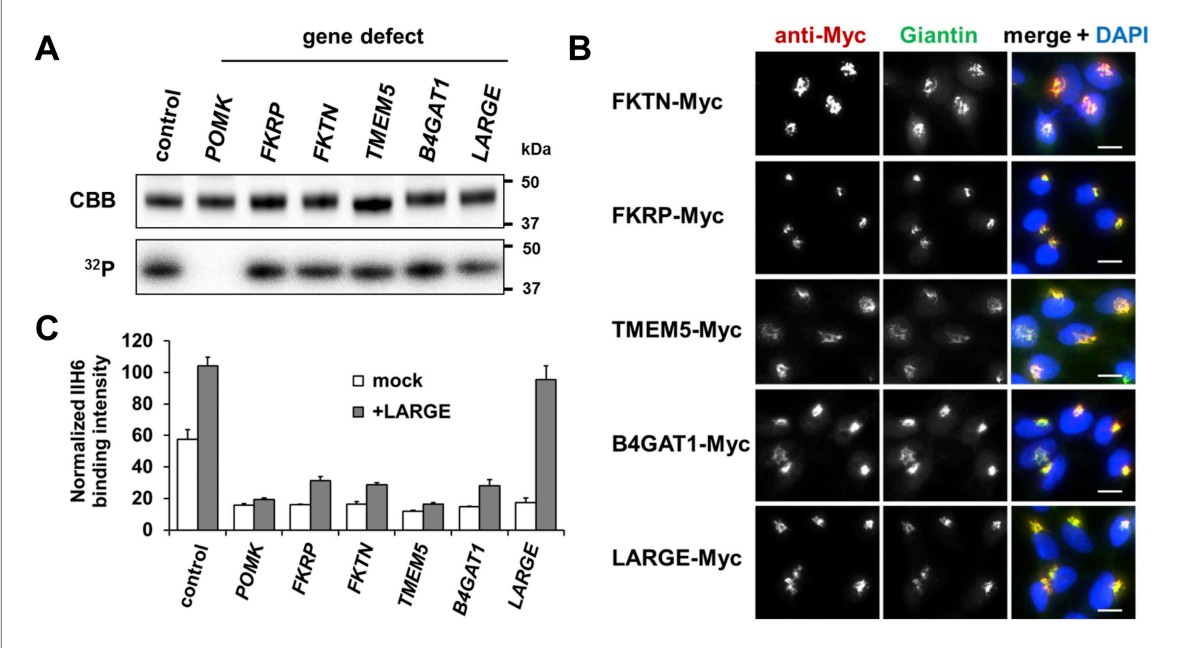

**Figure 1**. Postulated α-DG modifying enzymes are involved in post-phosphorylation processes in the Golgi prior to LARGE. (**A**) Phosphorylation of Fc-tagged DGFc340 in the context of α-DG glycosylation defects. Fc-tagged DGFc340 was produced in [$^{32}$P] orthophosphate-labeled fibroblasts from control and glycosylation-deficient patients and mice (*Table 1*). The DGFc340 was isolated from the culture medium by using protein A-agarose and the samples were separated by SDS PAGE. Gels were stained with Coomassie brilliant blue (CBB) and analyzed by phosphorimaging ($^{32}$P). (**B**) Subcellular localization of α-DG modifying putative glycosyltransferases, as assessed by immunofluorescence. HEK293T cells stably expressing c-Myc-tagged proteins were stained with anti-Myc (red), anti-Giantin (Golgi marker, green) and 4′,6-diamidino-2-phenylindole (DAPI, nuclei, blue). Individual stainings for c-Myc and Giantin are shown in greyscale and a merged image is shown in color. Scale bars indicate 10 μm. (**C**) Quantitative On-Cell protein blot analysis of LARGE-induced α-DG glycosylation hyperglycosylation in glycosylation-deficient cells. α-DG glycosylation status was tested with and without forced LARGE overexpression by adenovirus mediated gene transfer. The On-Cell Western blots were probed with an antibody against the glycosylated form of α-DG (IIH6). IIH6 On-Cell quantitative data were normalized with DRAQ5 cell DNA dye (n = 3). Error bars, SD.

The following figure supplement is available for figure 1:

**Figure supplement 1**. α-DG functional glycosylation and known proteins contributing to its synthesis.

(*Figure 2B*). However, in the presence of both UDP-Xyl and UDP-GlcA, the transfer of radioactivity was significantly increased consistent with the fact that the LARGE glycan is a heteropolysaccharide (*Figure 2A/B*). These results indicate that xylose is the initial sugar transferred by LARGE, and that this is followed by the transfer of GlcA to form the repeating [–3-xylose–α1,3-glucuronic acid-β1–] heteropolymer.

Next we wanted to identify the acceptor glycan used by LARGE to initiate formation of the laminin-binding glycan. Since the Xyl-T (xylosyltransferase) activity of LARGE has acceptor specificity for β-linked GlcA during heteropolymer formation, we hypothesized that β-linked GlcA might be the initial acceptor for the glycan added by LARGE. To test this, we pre-treated DGFc340 from *Large*$^{myd}$ MEF cells with β-glucuronidase (β-GUS) (*Figure 2A/C*), and assessed its modification by LARGEdTM in an in vitro assay. Subsequent immunoblotting with the LARGE glycan-specific antibody (IIH6) revealed that the pretreatment of *Large*$^{myd}$ DGFc340 with β-glucuronidase resulted in a strong reduction of the IIH6 signal (*Figure 2C*). These data indicate that LARGE uses a β-linked GlcA residue as an acceptor sugar to initiate synthesis of the polymeric glycan.

## Xylose is part of the α-DG O-mannosyl post-phosphoryl modification

To determine which monosacharides contribute to synthesis of the O-mannosyl post-phosphoryl acceptor for the LARGE glycan, we performed a LARGEdTM assay with DGFc340 acceptor isolated from a panel of sugar nucleotide-deficient CHO (Chinese hamster ovary) cells (*Stanley, 1985*; *Kingsley et al., 1986*). LARGEdTM was able to efficiently modify DGFc340 from Pro5 (wild-type),

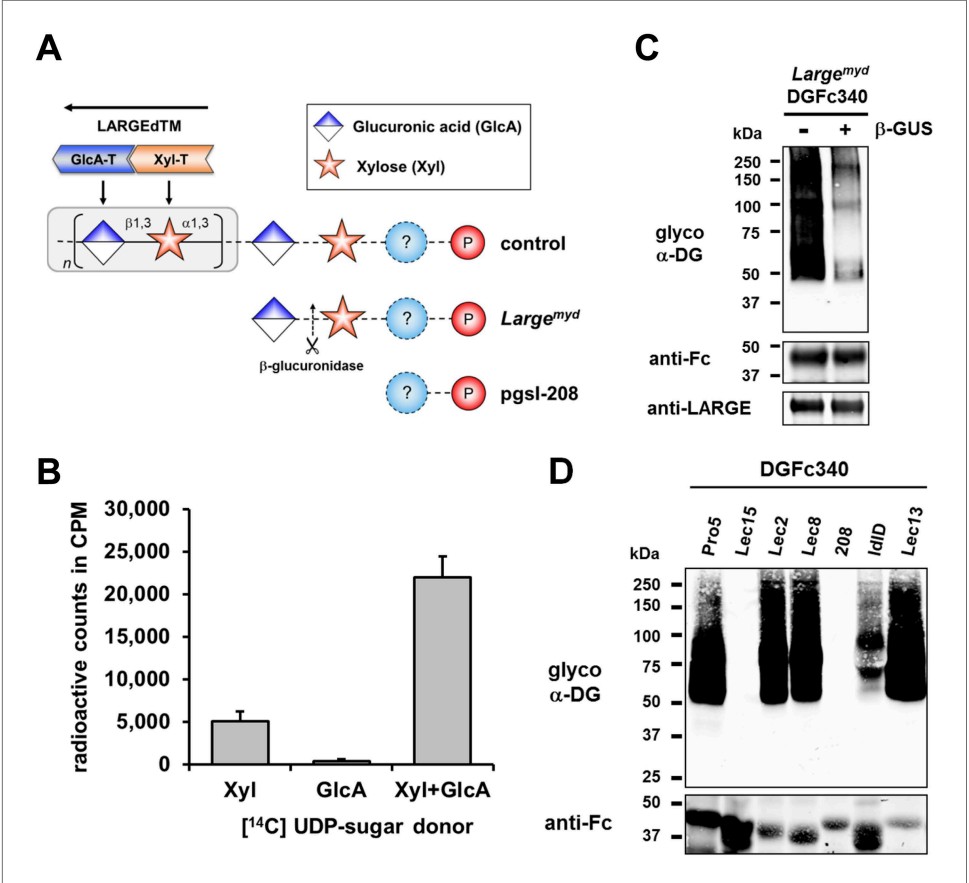

**Figure 2**. β-GlcA serves as an acceptor sugar for LARGE modification starting with xylose. (**A**) Schematic diagram showing the α-DG post-phosphoryl modification in the context of control and glycosylation defects. LARGE adds the ligand-binding glycan to α-DG via a proposed glucuronic acid (GlcA) acceptor. LARGEdTM catalytic domains Xyl-T (orange) and GlcA-T (blue) are highlighted in color. Depicted are also the hypothesized terminal sugar structures of glycosylation-deficient cell lines *Large^{myd}* (*Large*-deficient) and pgsl-208 (UDP-xylose deficient). Cleavage of terminal β-GlcA by exoglycosidase β-glucuronidase (β-GUS) in *Large^{myd}* is indicated (scissor symbol). (**B**) Transfer of [14C] radiolabeled Xyl and GlcA to DGFc340 by LARGEdTM. Fc-tagged DGFc340 was produced in *Large^{myd}* (*Large*-deficient) MEF cells and isolated from the culture medium using protein A-agarose. The protein A-bound DGFc340 was used as acceptor in a LARGEdTM reaction with radiolabeled [14C] UDP-Xyl and/or [14C] UDP-GlcA sugar donors. The figure represents the transfer of radiolabeled saccharides onto the donor DGFc340 (n = 3). Error bars represent SD (**C**) β-Glucuronidase pre-treatment of DGFc340 from *Large^{myd}* deficient cells impairs LARGEdTM modification. Protein A-bound DGFc340 (acceptor) isolated from transfected *Large^{myd}* MEFs was digested with β-glucuronidase (β-GUS) prior to the LARGEdTM (enzyme) reaction, which included UDP-Xyl and UDP-GlcA as sugar (donors). After incubation with LARGEdTM DGFc340 (acceptor protein) was subjected to protein blotting with antibodies against the glycosylated form of α-DG (IIH6), against Fc and against LARGE (Rb331). (**D**) The ability of LARGEdTM to modify DGFc340 is impaired in the context of sugar donor-deficient CHO mutant cell lines. Fc-tagged DGFc340 was produced in various glycosylation-deficient Lec CHO cells and isolated from the culture medium using protein A-agarose. As in (**C**) protein A-bound DGFc340 acceptor was used in a LARGEdTM reaction and analyzed by protein blotting.

Lec2 (CMP-sialic acid-deficient), Lec8 (UDP-galactose-deficient) and Lec13 (GDP-fucose-deficient) cells, suggesting that sialic acid, galactose and fucose do not contribute to functional glycosylation of α-DG (*Figure 2D*). ldlD cells deficient for UDP-galactose (UDP-Gal) and UDP-N-acetylgalactosamine (UDP-GalNAc) demonstrated reduced acceptor activity for LARGEdTM, which can be explained by the fact that B3GALNT2 requires UDP-GalNAc for synthesis of the initial Core M3 structure. Similarly, DGFc340 from Lec15 cells, which are deficient for Dol-P-Man synthesis, did not serve as LARGEdTM acceptor because POMT1 (Protein *O*-mannosyltransferase) and POMT2 require the Dol-P-Man sugar

donor to initiate the O-mannosyl Core M3 structure. Most interestingly, in the LARGEdTM in vitro assay DGFc340 from UDP-Xyl-deficient pgsl-208 CHO cells was not modified (*Figure 2D*). It had also been shown that ectopic LARGE expression in pgsl-208 CHO cells did not induce α-DG hyperglycosylation (*Inamori et al., 2012*; *Ashikov et al., 2013*), consistent with the fact that UDP-Xyl is essential for synthesis of the LARGE glycan in vivo. However, in our LARGEdTM in vitro assay, pgsl-208 DGFc340 was also not modified by LARGEdTM, despite the presence of both of the required sugar nucleotides, UDP-GlcA and UDP-Xyl. This clearly demonstrated that one or more xylose residues are required on α-DG before it can be functionally glycosylated by LARGE (*Figure 2A*).

## B4GAT1 is a glucuronyltransferase with specificity for the β-xylose acceptor

Having identified a β-linked glucuronic acid as the terminal acceptor saccharide for LARGE and xylose as component of the α-DG post-phosphoryl glycan modification, we next sought to determine which enzyme is responsible for the hypothesized glucuronyltransferase activity. Among the group of unassigned genes (*FKTN, FKRP, TMEM5, B4GAT1*) only the *B4GAT1* gene product showed homology to glucuronyl-transferases. In particular it shares 44% similarity with the LARGE GlcA-T (Glucuronyltransferase) domain (CAZy: GT49; *Figure 3A*). This designated B4GAT1 as a promising

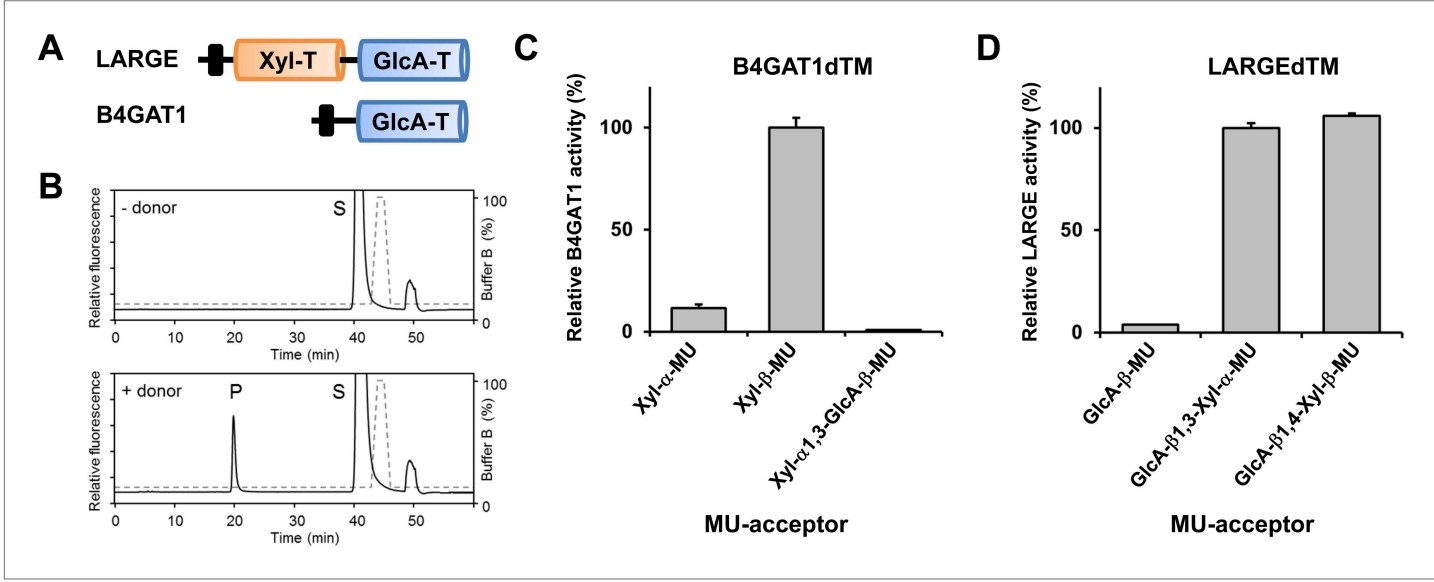

**Figure 3**. B4GAT1 has xylose β1,4 glucuronyltransferase activity. (**A**) Schematic representation of LARGE and B4GAT1 functional domains. GlcA-T (blue), Xyl-T (orange) and transmembrane domain (black) are indicated. (**B**) Representative HPLC profiles of the reaction product generated in the absence (top) and presence (bottom) of a UDP-GlcA sugar (donor) in a reaction mix containing Xyl-β-MU (acceptor) and B4GAT1dTM (enzyme). Samples were separated on an LC-18 column. P, product. S, unreacted substrate. Dotted line, %B buffer. (**C**) Comparison of B4GAT1dTM GlcA-T activity with respect to various xylose-MU acceptor sugars. Relative activity (%) with respect to Xyl-β-MU acceptor (specific activity: 0.2 μmol/h/mg) is shown (n = 3). Error bars represent SD. (**D**) Comparison of LARGEdTM Xyl-T activity with respect to various monosaccharide and disaccharide GlcA-MU acceptor sugars. Relative activity (%) with respect to intrinsic LARGE polymer specific activity with GlcA-β1,3-Xyl-α-MU disaccharide acceptor (0.08 μmol/h/mg) (n = 3). Error bars represent SD.

The following source data and figure supplements are available for figure 3:

**Source data 1**.

**Figure supplement 1**. Purification of B4GAT1dTM.

**Figure supplement 2**. Basic characterization of the xylose β1,4-glucuronyltransferase activity of B4GAT1.

**Figure supplement 3**. NMR analysis reveals that B4GAT1 is a β1,4 glucuronyltransferase.

**Figure supplement 4**. Test B4GAT1 for GlcNAc transferase activity with iGnT substrate Gal-β1,4-GlcNAc-β-MU.

candidate for the GlcA-T transferase upstream of LARGE. To test this hypothesis, we generated a 6xHis-tagged soluble construct of B4GAT1 (transmembrane domain deleted, B4GAT1dTM), expressed it in HEK293T cells and purified the recombinant enzyme from the culture medium (*Figure 3—figure supplement 1*). We then conducted a transfer assay with B4GAT1dTM as the enzyme source, UDP-GlcA as the sugar donor and fluorescently labeled β-xyloside (4-methylumbelliferyl-β-D-xyloside, Xyl-β-MU) as the acceptor. The reaction products were separated by high-performance liquid chromatography (HPLC). A unique product peak was detected only when UDP-GlcA was used as donor (*Figure 3B*, *Figure 3—figure supplement 2A*). We also tested the acceptor specificity, which revealed that B4GAT1dTM GlcA-T activity has low preference for α-linked Xyl, but showed >10- fold higher preference and specificity towards β-linked Xyl acceptors (*Figure 3C*, *Figure 3—figure supplement 2A*). The fact that the LARGE glycan disaccharide Xyl-α1,3-GlcA-MU was a very weak acceptor for B4GAT1dTM GlcA-transfer suggests that B4GAT1 overexpression does not interfere with LARGE mediated synthesis of the laminin-binding glycan. A characterization of the B4GAT1 GlcA-T activity revealed a metal dependence for manganese ($Mn^{2+}$) divalent cations (*Figure 3—figure supplement 2B*) and a pH-optimum near pH 7.0 (*Figure 3—figure supplement 2C*). The product peak obtained from the enzymatic reaction of B4GAT1dTM with β-Xyl-MU acceptor was isolated, and its analysis by NMR revealed that the GlcA residue was β-linked to the four position of the xylose β-MU (*Figure 3—figure supplement 3*, *Figure 3—source data 1*). Thus, B4GAT1 possesses xylose β1,4-glucuronyltransferase (GlcA-T) activity and it is specific for the substrate β-linked Xyl.

## The substrate specificity of LARGE Xyl-T is not dependent on the glycosidic bond of the β-GlcA acceptor

It had previously been shown that, during synthesis of the LARGE heteropolysaccharide, β1,3-linked GlcA serves as the acceptor for LARGE Xyl-T (*Inamori et al., 2012*). In the current study we found that a β1,4-linked GlcA transferred by B4GAT1 serves as the acceptor glycan for initiation of synthesis of the LARGE glycan, via the addition of a xylose. To assess if LARGE can use one or the other glycosidic linkage β-linked GlcA acceptor with higher efficiency, we tested LARGEdTM Xyl-T activity on two disaccharides GlcA-β1,4-Xyl-β-MU and GlcA-β1,3-Xyl-α-MU along with the monosaccharide GlcA-β-MU (4-methylumbelliferyl-β-D-glucuronide). As shown in *Figure 3D*, LARGE did not distinguish between GlcA-β1,3-Xyl and GlcA-β1,4-Xyl, as similar activities were measured in the presence of both disaccharide acceptors. However, the length of the acceptor appears to be important, since the disaccharide acceptor showed 25-fold higher activity than the monosaccharide acceptor (*Figure 3D*). In summary, it is currently unknown why the β1,4-linked GlcA acceptor in the initial LARGE acceptor primer and the β1,3-linked GlcA in the terminal LARGE glycan have different linkages, and how each linkage contributes spatially to the overall structure, while LARGE shows similar activity towards both acceptors.

## MEFs from *B4gat1*-deficient mice lack endogenous B4GAT1 activity

To elucidate further the role of B4GAT1 in vivo, we isolated MEFs from control, *Large^myd* (*Large*-deficient) and *B4gat1*-deficient mice (*Wright et al., 2012*) (*Table 1*) and analyzed the glycosylation status of α-DG. Immunoblotting revealed that whereas control MEFs were positive for functional glycosylation of, and laminin-binding by, α-DG from *Large^myd* MEFs completely lacked both features (*Figure 4A*). Also, *B4gat1*-deficient MEFs demonstrated strongly reduced but detectable residual functional glycosylation and laminin binding, and normal levels of hypoglycosylated α-DG core protein (*Figure 4A*). Adenovirus-mediated ectopic expression of B4GAT1 did not affect the glycosylation status of α-DG in control and *Large^myd* MEFs but, as expected, was able to rescue the α-DG glycosylation defect in *B4gat1*-deficient MEFs (*Figure 4A*). As demonstrated previously (*Barresi et al., 2004*; *Inamori et al., 2012*; *Willer et al., 2012*), forced adenovirus-mediated ectopic expression of LARGE in control and *Large^myd* MEFs induces α-DG hyperglycosylation. In contrast, *B4gat1*-deficient cells showed only a low level of α-DG hyperglycosylation after LARGE overexpression (*Figure 4A*), suggesting that in the context of mutant B4GAT1, only few acceptor sites for LARGE modification are available. Finally, ectopic co-expression of B4GAT1 and LARGE resulted in α-DG hyperglycosylation in all three tested MEF lines (*Figure 4A*). This result is consistent with the hypothesis that B4GAT1 acts prior to the glycosyltransferase LARGE.

Next, to determine if endogenous B4GAT1 activity was detectable in MEF cells, we subjected samples from control, *B4gat1* and *Large^myd* MEFs to the B4GAT1 enzyme activity assay, using Xyl-β-MU as the acceptor. Whereas control and *Large^myd* samples showed comparable B4GAT1 transferase activity, only low residual activity (<3%) was detectable in *B4gat1*-deficient cells, and this loss could be restored

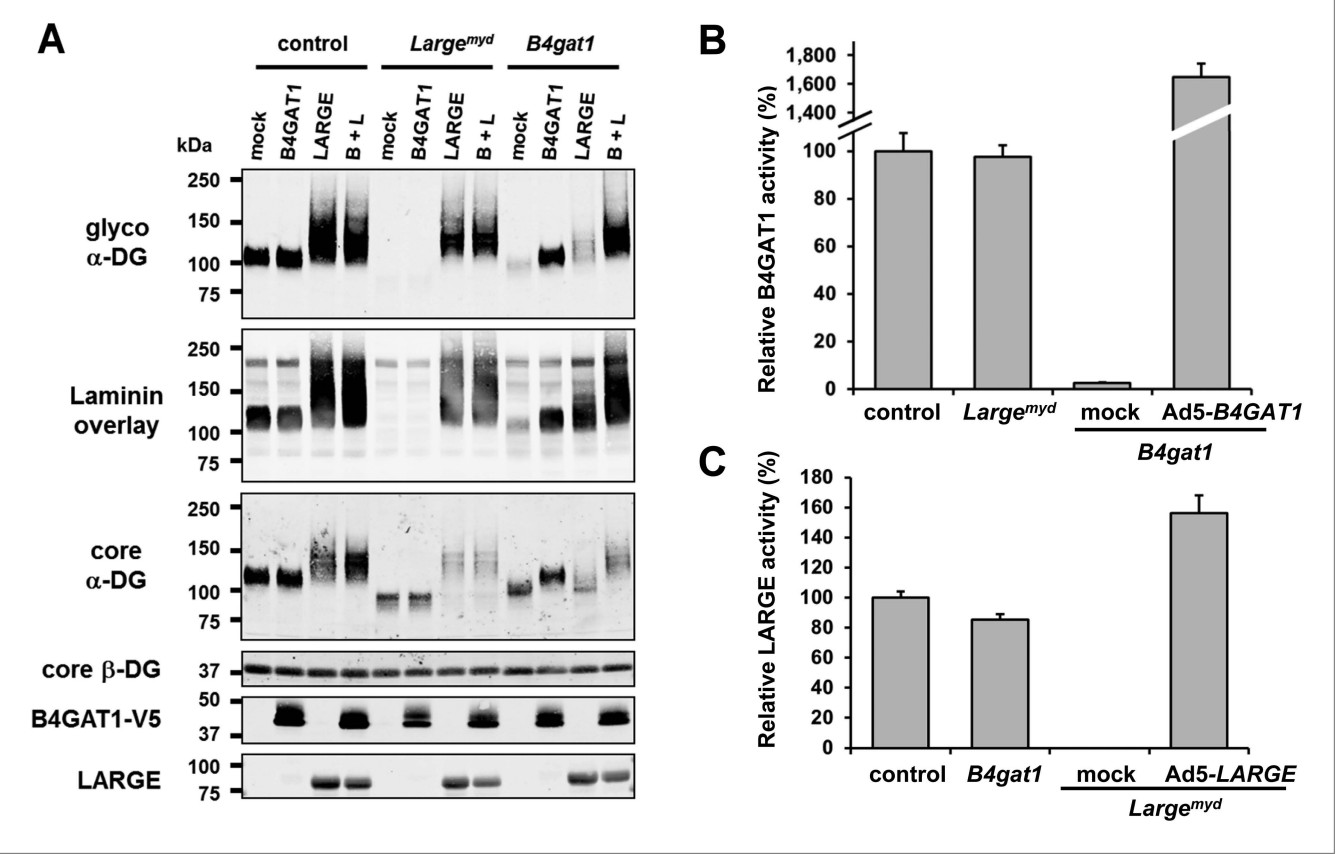

**Figure 4**. *B4gat1*-deficient MEFs have impaired α-DG functional glycosylation and endogenous B4GAT1 activity. (**A**) Functional glycosylation and complementation analysis of α-DG in wild-type, *Large*- and *B4gat1*-deficient MEFs. Immunoblots and laminin overlay assay of WGA-enriched cell lysates extracted from WT, *Large^myd* (*Large*-deficient) and *B4gat1*-deficient MEFs . As indicated MEFs were uninfected (mock) or infected with adenovirus constructs expressing B4GAT1, LARGE or both (B + L). Antibodies used were: glyco α-DG (IIH6), core α-DG, core β-DG (AP83), anti-V5 and anti-LARGE (Rb331). (**B**) Comparison of endogenous B4GAT1 GlcA-T activity in control, *Large*- and *B4gat1*-deficient MEFs. Additionally, *B4gat1*-deficient MEFs (*B4gat1^{LacZ/M155T}*) complemented with control B4GAT1 expressing adenovirus (Ad5) were tested. Cell lysates were used as enzyme source to measure endogenous B4GAT1 activity. Relative activity (%) with respect to control MEFs specific activity (91.6 pmol/h/mg) is shown (n = 3). Error bars represent SD. (**C**) Comparison of endogenous LARGE GlcA-T activity in control, *Large*- and *B4gat1*-deficient MEFs. Additionally, *Large*-deficient MEFs complemented with control *LARGE* expressing adenovirus (Ad5) were tested. WGA enriched glycoprotein samples were used as enzyme source to measure endogenous LARGE activity. Relative activity (%) with respect to control MEFs specific activity (0.52 pmol/h/mg) is shown (n = 3). Error bars represent SD.

The following figure supplement is available for figure 4:

**Figure supplement 1**. B4gat1-deficient MEFs have impaired endogenous B4GAT1 activity.

by ectopic expression of B4GAT1 (*Figure 4B*, *Figure 4—figure supplement 1A/B*). Similarly, when we tested LARGE GlcA-T activity in control and glycosylation-deficient MEFs, only *Large^myd* MEFs lacked LARGE GlcA-T activity; the control and *B4gat1*-deficient cells were normal (*Figure 4C*, *Figure 4—figure supplement 1C/D*). These results suggest that the enzymatic activities of B4GAT1 and LARGE are independent and that each is unaffected by mutations in the gene product of the other.

## B4GAT1 mutations affect the subcellular localization and activity of B4GAT1

To date, several disease-causing *B4GAT1* (formerly termed *B3GNT1*) mutations have been reported in human patients (*Buysse et al., 2013*; *Shaheen et al., 2013*), in an N-ethyl-N-nitrosourea (ENU)-induced mutant mouse model (*Wright et al., 2012*) and in a genetic screen for modifiers of LASV entry (*Jae et al., 2013*). To test how reported *B4GAT1* missense mutations affect the intracellular localization of B4GAT1 and its enzymatic activity, we cloned three mutant B4GAT1-Myc expression constructs (*Figure 5A*): Mut1 (N390D) represents a mutation identified in a patient with Walker-Warburg

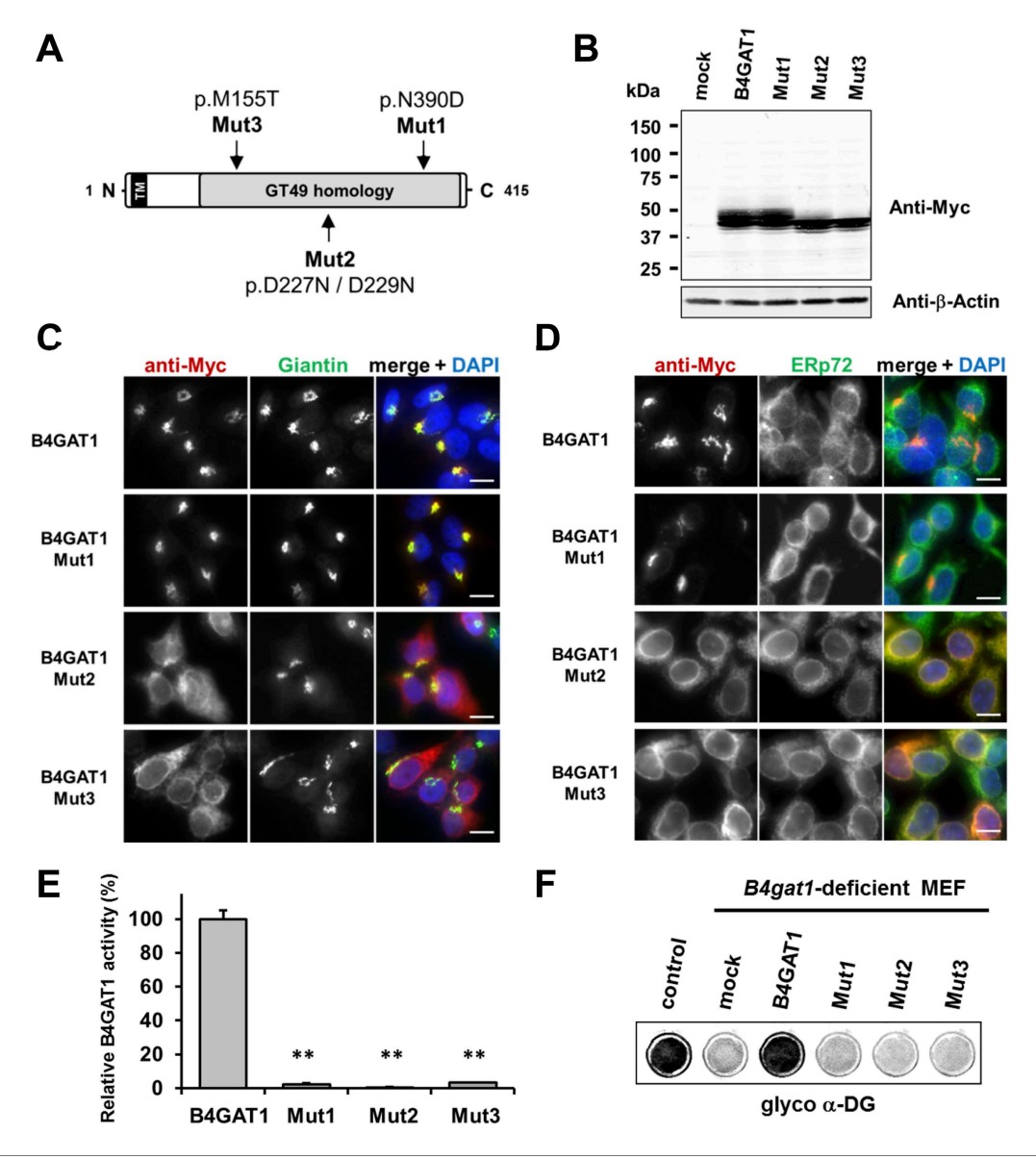

**Figure 5**. Expression analysis and GlcA-T enzyme activity of B4GAT1 mutant constructs. (**A**) Schematic presentation shows B4GAT1 enzyme product with functional domains and B4GAT1 mutations Mut1-Mut3 are indicated. (**B**) Expression analysis of B4GAT1-Myc control and mutant constructs in stable HEK293T cells. Immunoblotting of cell lysates from HEK293T cells stably overexpressing wild-type B4GAT1-Myc and mutant constructs (Mut1, Mut2 and Mut3) with anti-Myc antibody and β-Actin (loading control). (**C/D**) Subcellular localization of B4GAT1-Myc control and mutant constructs in stable HEK293T cells (see **B**). B4GAT1-Myc constructs were stained with anti-Myc (red), (**C**) anti-Giantin (Golgi marker, green), (**D**) anti-ERp72 (ER marker, green) and 4′,6-diamidino-2-phenylindole (DAPI, nuclei, blue). Individual stainings for c-Myc Giantin and ERp72 are shown in greyscale, and merged images are shown in color. Scale bars indicate 10 μm. (**E**) B4GAT1 enzyme activity in cell lysates from stable HEK293T cells overexpressing B4GAT1-Myc wild-type and B4GAT1-Myc mutant constructs (Mut1-Mut3). Relative activity (%) with respect to B4GAT1wild-type specific activity (19.8 nmol/hr/mg) is shown (n = 3). Error bars represent SD, Statistical analyses were performed by two-tail Student's $t$ test. **p < 0.001. (**F**) Complementation of *B4gat1*-deficient (*B4gat1^LacZ/M155T*) MEF cells with B4GAT1-Myc control and mutant constructs. *B4gat1*-deficient MEFs were nucleofected with a wild-type or mutant B4GAT1 expression construct. α-DG functional glycosylation was analyzed by On-Cell-Western analysis. α-DG functional glycosylation was detected with α-DG glyco (IIH6) antibody.

syndrome (**Buysse et al., 2013**); Mut2 (D227N/D229N) is a mutation in the glycosyltransferase signature DXD motif (**Wiggins and Munro, 1998**; **Bao et al., 2009**); and Mut3 (M155T) mimics a mutant allele identified in a *B4gat1*-deficient mouse model with axon guidance defects (**Wright et al., 2012**). Immunoblot analysis confirmed that expression levels were similar for the Myc-tagged B4GAT1 control and all three mutant constructs Mut1-Mut3 in stably expressing HEK293T cells (**Figure 5B**). Previously, it had been shown that B4GAT1 localizes to the *trans*-Golgi near the TGN (*trans*-Golgi network) (**Bao et al., 2009**; **Lee et al., 2009**; **Buysse et al., 2013**). In our immunofluorescence analysis, we found both the control and mutant construct Mut1 to exhibit normal Golgi localization and to co-localize with the Golgi marker Giantin (**Figure 5C**). B4GAT1 mutations in constructs Mut2 and Mut3, however, resulted in a high degree of mislocalization to the ER, as judged by overlap of the signal with the ER marker ERp72 (**Figure 5D**), indicating that the B4GAT1 mutant proteins are misfolded and retained in the ER. Analysis of B4GAT1 enzyme activity in lysates from cells stably overexpressing these constructs revealed strongly reduced activity in the cases of all three mutants, of less than 5% compared to activity levels in wild-type control (**Figure 5E**). Similarly, none of the B4GAT1 mutant constructs was able to complement and rescue the α-DG glycosylation defect in *B4gat1*-deficient MEFs (**Figure 5F**). These findings confirm that the identified B4GAT1 mutations are pathological and have a direct negative impact on B4GAT1 activity regardless of their subcellular localization. Additionally, the finding that the B4GAT1 DXD motif is essential further supports a role for B4GAT1 as a glycosyltransferase, since the DXD motif is thought to be involved in binding carbohydrate sugar-nucleoside diphosphates in manganese-dependent glycosyltransferases (**Wiggins and Munro, 1998**).

## β-Xylose serves as an endogenous acceptor for B4GAT1

To further characterize the endogenous acceptor for B4GAT1, we first tested if B4GAT1dTM was able to use DGFc340 from control, *Large^myd* and *B4gat1*-deficient MEFs as an acceptor. Similar to the LARGE acceptor experiment in **Figure 2B**, we used radiolabeled [$^{14}$C] UDP-GlcA sugar donor and measured transfer of [$^{14}$C] to the protein A-bound DGFc340 acceptor. As expected, DGFc340 isolated from *B4gat1*-deficient cells was the only acceptor that incorporated substantial levels of the radioactive label (**Figure 6A**). This confirmed that only the DGFc340 acceptor from *B4gat1*-deficient cells resembled the terminal acceptor glycan suitable for B4GAT1dTM to add GlcA. Our B4GAT1dTM in vitro enzyme assay demonstrated acceptor specificity for β-linked xylose (**Figure 3C**). To corroborate the hypothesis that β-linked xylose also serves as the endogenous B4GAT1 α-DG acceptor we

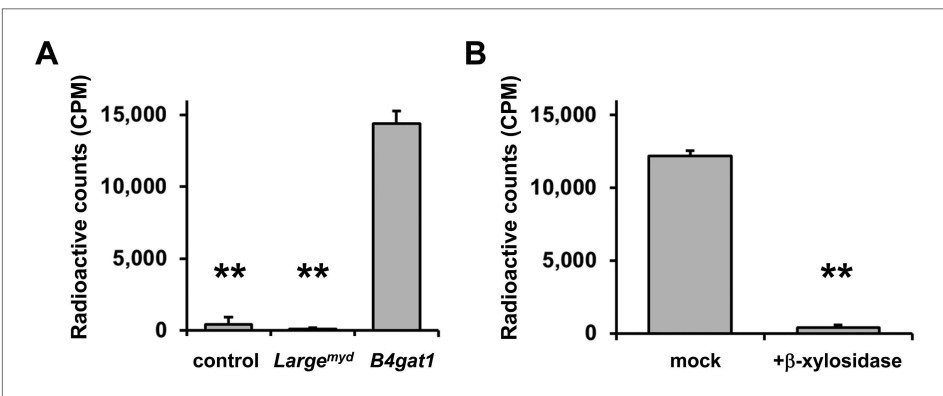

**Figure 6**. β-xylose is the endogenous acceptor for B4GAT1. (**A**) B4GAT1dTM enzymatic transfer of [$^{14}$C] radiolabeled GlcA to DGFc340. Fc-tagged DGFc340 (acceptor) was produced in control, *Large^myd* (*Large*-deficient) and *B4gat1*-deficient MEFs and isolated from the culture medium using protein A-agarose. The protein A-bound Fc340 was used as acceptor in a B4GAT1dTM (enzyme) reaction with radiolabeled [$^{14}$C] UDP-GlcA sugar (donor). The figure represents the transfer of radiolabeled GlcA onto the donor DGFc340 (n = 3). Error bars represent SD. Statistical analyses were performed by two-tail Student's *t* test. **p < 0.001. (**B**) β-Xylosidase pre-treatment impairs B4GAT1dTM transfer of [$^{14}$C] radiolabeled GlcA. DGFc340 (acceptor) from *B4gat1*-deficient MEFs was digested with β-xylosidase prior to the B4GAT1dTM (enzyme) transfer reaction with [$^{14}$C] UDP-GlcA sugar (donor). The figure represents the transfer of radiolabeled GlcA onto the donor DGFc340 (n = 3). Error bars represent SD Statistical analyses were performed by two-tail Student's *t* test. **p < 0.001.

pre-treated DGFc340 from *B4gat1*-deficient cells with β-xylosidase and measured the transfer of [$^{14}$C] GlcA by B4GAT1dTM. After β-xylosidase treatment, the ability of *B4gat1*-deficient DGFc340 to act as an acceptor was strongly reduced; this constitutes indirect evidence that a β-linked xylose is indeed the postulated endogenous acceptor for B4GAT1 (*Figure 6B*), and that a yet unidentified xylosyltransferase acts upstream of B4GAT1.

## NMR-studies confirm that B4GAT1 synthesizes the acceptor glycan for LARGE

To further corroborate our finding that the glycosyltransferase LARGE utilizes a glucuronic acid-β1,4-xylose-β disaccharide acceptor as a primer to elongate it with its dual glycosyltransferase and polymerizing activity, we performed NMR structural studies. In our approach towards confirming each individual glycosidic linkage, we first synthesized the tetrasacharide GlcA-Xyl-GlcA-Xyl-MU, starting with the monosaccharide acceptor Xyl-β-MU and extending it in a stepwise manner using recombinant B4GAT1dTM and LARGEdTM as enzymes sources (*Figure 7A*).

The $^1$H and $^{13}$C resonances of the isolated tetrasaccharide product were assigned by using heteronuclear multiple quantum coherence (HMQC), heteronuclear 2-bond correlation (H2BC), and heteronuclear multiple bond coherence (HMBC) spectra (*Figure 7B*, *Figure 7—source data 1*). The detection of the interglycosidic cross-peaks of BH1/AC4, CC1/BH3, and DH1/CC3 in the HMBC spectrum (*Figure 7B*) clearly indicates the presence of a 1→4 interglycosidic linkage between sugar residues B and A, a 1→3 interglycosidic linkage between residues C and B, and a 1→3 interglycosidic linkage between residues D and C, respectively. A strong rotating-frame Overhauser effect (ROE) was observed from the H1 to H3 and H5 protons of residues A, B, and D in the ROE spectroscopy (ROESY) spectrum (*Figure 7C*), demonstrating that they have a β-configuration. The observed strong ROE from the residue C H1 proton to its own H2, but not to H3 and/or H5 demonstrates that the residue C has an α-configuration. The inter-residue ROEs observed in the ROESY spectrum are also consistent with the interglycosidic linkage assignments determined from the HMBC spectrum. Therefore, the tetrasaccharide has the glycosidic linkage structure GlcA-β1,3-Xyl-α1,3-GlcA-β1,4-Xyl-β-MU (*Figure 7A*). These studies show that B4GAT1 possesses β1,4 glucuronyltransferase activity, and that LARGE can elongate this primer structure by adding repeating units [-3-Xyl-α1,3-GlcA-β1-] to produce a heteropolysaccharide. To further illustrate the complexity of assembling the functional glycan of α-DG, we summarize the current knowledge about the α-DG sugar structures and the contributing genes/enzymes in *Figure 8*.

## Discussion

In this study we used a multidisciplinary approach to investigate how the assembly of the α-DG LARGE glycan is initiated, and found that it requires B4GAT1-dependent synthesis of a novel glucuronyl-xylosyl acceptor primer. We show that B4GAT1 is a xylose β1,4-glucuronyltransferase, and that it is involved in synthesizing the glycan primer that subsequently can be elongated by LARGE with the ligand-binding glycan. B4GAT1 was initially cloned and described by Sasaki et al., (*Sasaki et al., 1997*) as β1,3-N-acetylglucosaminyltransferase (iGnT, β3GNT1 or B3GNT1), which is essential for the synthesis of poly-N-acetyllactosamine. Furthermore, the B3GNT1 enzyme was proposed to contribute to the i antigen synthesis pathway by transferring N-acetylglucosamine onto a β-galactose acceptor with β1,3 linkage (*Sasaki et al., 1997*). In contrast our data reveal a β1,4-glucuronyltransferase activity, which we have designated B4GAT1. We tested B4GAT1dTM with UDP-GlcNAc and the proposed Gal-β1,4-GlcNAc-β-MU acceptor, but we were not able to validate any N-acetylglucosaminyltransferase activity (*Figure 3—figure supplement 4*). Therefore, we propose to rename the enzyme B4GAT1, as a new member of the glucuronyltransferase family of proteins. To date, only 2 other enzymes are known to have β1,4-glucuronyltransferase activity. These are EXT1 and EXT2, and both are involved in the synthesis of heparan sulphate proteoglycans (*Lidholt and Lindahl, 1992*).

Our findings regarding assembly of the LARGE glycan reveal striking similarities to the unique mechanism underlying the synthesis of proteoglycans. Both glycan polymers consist of repeating disaccharides that are synthesized by glycosyltransferases with dual glycosyltransferase activities (*Esko et al., 2009*; *Inamori et al., 2012*). Furthermore, in both cases assembly of the terminal heteropolymer glycan is initiated by a disaccharide primer, which is part of a larger glycan linker that anchors the polysaccharide to a protein backbone. Future studies are needed to elucidate the full α-DG glycan structure and determine the roles of the putative glycosyltransferases FKTN, FKRP and TMEM5 in anchoring the α-DG ligand-binding glycan moiety to the phosphorylated Core M3 structure. At this

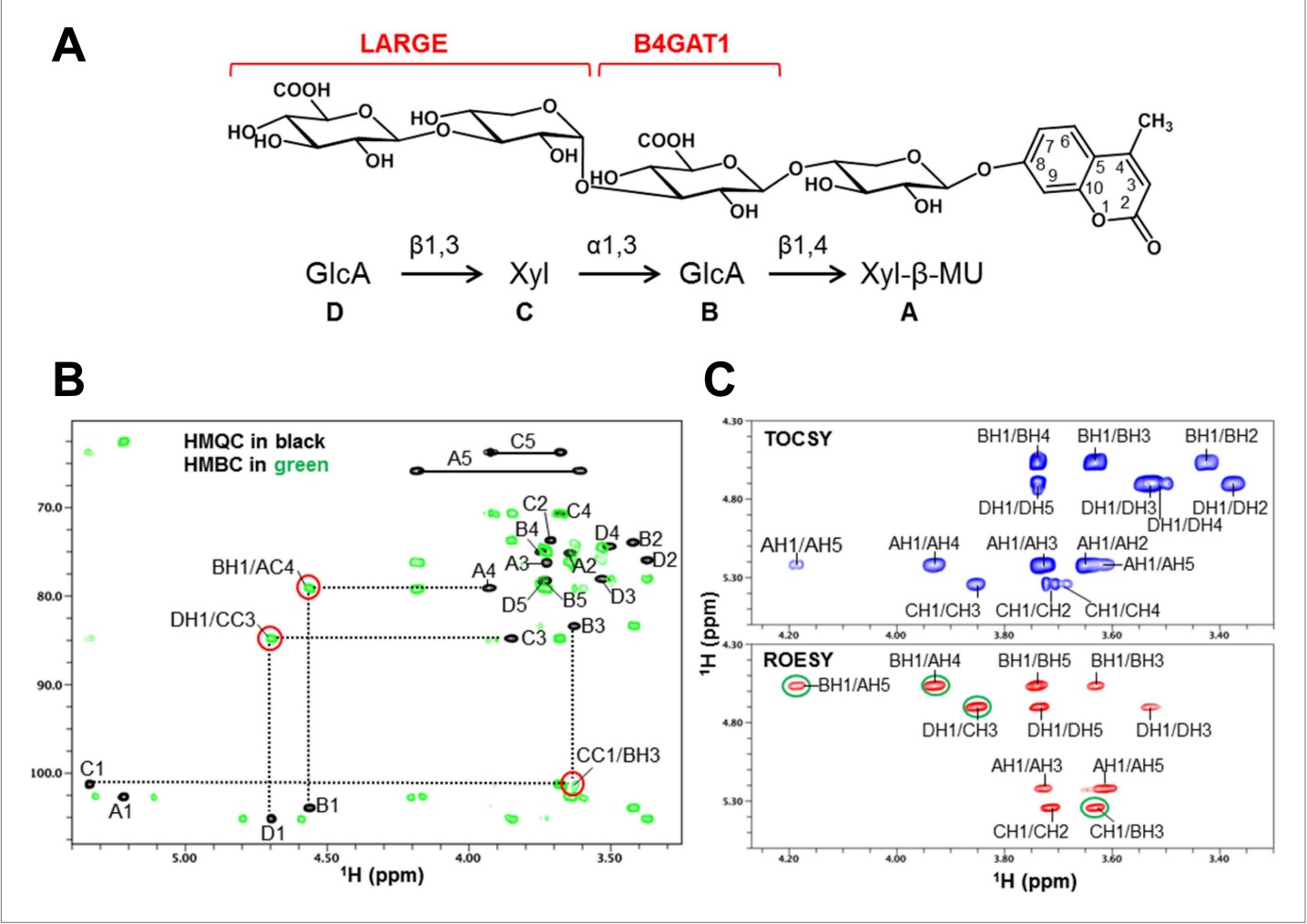

**Figure 7**. NMR analyses of the tetrasacharide generated from B4GAT1dTM and LARGEdTM enzymatic reactions. (**A**) Schematic depiction of the tetrasaccharide structure produced by the sequential reactions of B4GAT1dTM followed by LARGEdTM with the sugar units labeled A-D to indicate the order of their addition. (**B**) Overlay of the HMQC (black) and HMBC (green) spectra of the tetrasaccharide. All cross-peaks in the HMQC spectrum are labeled. Three interglycosidic cross-peaks detected in the HMBC spectrum are also labeled and indicated with red circles. The peaks are labeled with a first letter representing the subunit designated in **A,** and the rest of the label representing the position on that subunit. (**C**) TOCSY spectrum (top) and ROESY spectrum (bottom) of the tetrasaccharide. The TOCSY and ROESY spectra were collected with mixing time of 77 and 300 ms, respectively. The cross-peaks are labeled as in **B**. The observed interglycosidic ROEs are indicated with green circles.

The following source data is available for figure 7:

**Source data 1**.

point it cannot be ruled out that other, currently unidentified, genes also contribute to synthesis of the functional α-DG glycan.

Similar to their counterparts in other dystroglycanopathy genes, *B4GAT1 (B3GNT1)* loss-of-function mutations in human patients result in Walker-Warburg Syndrome (WWS) (*Buysse et al., 2013*; *Shaheen et al., 2013*), the most severe condition in a range of clinically defined CMDs that are accompanied by brain and eye malformations. Milder *B4GAT1* mutations with residual enzyme activity are expected to cause a milder Limb Girdle Muscular Dystrophy (LGMD) phenotype, but patients with such mutations have not yet been described.

*B4gat1 (B3gnt1)*-null mutations in mice result in early embryonic lethality, at ~ E9.5 (*Wright et al., 2012*), as is the case for reported null mutations in *Dag1* (*Williamson et al., 1997*), *Pomt1* (*Willer et al., 2004*), and *Fukutin* (*Kurahashi et al., 2005*). Proper α-DG glycosylation is essential for early

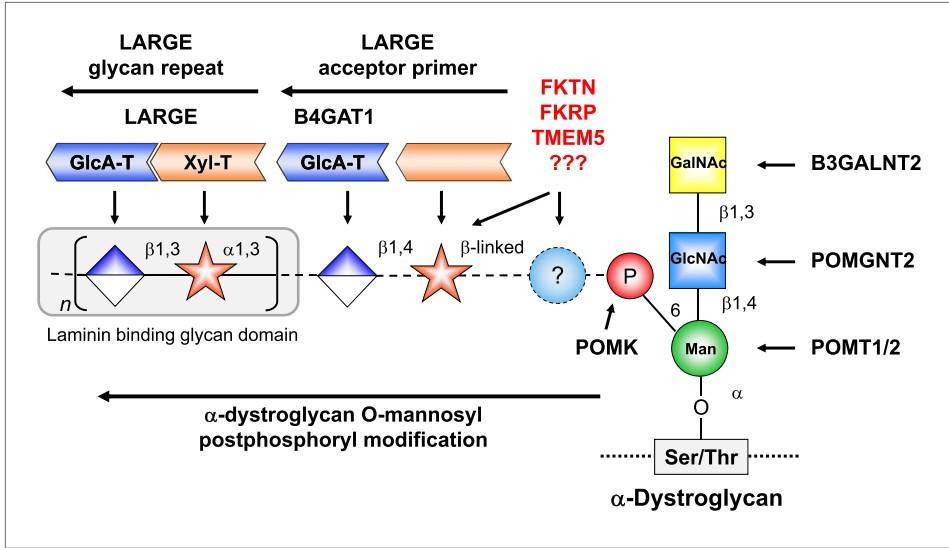

**Figure 8**. Model of proposed α-DG O-mannosyl laminin-binding glycan structure and the enzymes that contribute to its synthesis. Post-phosphoryl modification of α-DG requires B4GAT1 (β1,4 glucuronyltransferase); this enzyme generates the acceptor glycan, which serves as a primer for the glycosyltransferase LARGE to initiate synthesis of the laminin-binding glycan. Both gene products with known function (black) and gene products with currently unidentified function (red) are indicated.
The following figure supplement is available for figure 8:

**Figure supplement 1**. B4GAT1 *and LARGE* expression in human tissues.

embryonic development in the mouse, including formation of the basement membrane, as defects in the Reichert's membrane are the suspected cause of death in α-DG glycosylation-deficient mice (**Williamson et al., 1997**; **Willer et al., 2004**; **Kurahashi et al., 2005**). However, an ENU-based genetic screen for abnormal CNS axonal tracks identified a viable *B4gat1 (B3gnt1)* dystroglycanopathy mouse model carrying a p.M155T *B4gat1* mutation (**Wright et al., 2012**). The majority of compound hetero-zygous mice with both a *LacZ* (*B4gat1^{LacZ}*) null allele and a hypomorphic p.M155T (*B4gat1^{M155T}*) allele die perinatally, but a few survive and develop a characteristic CMD phenotype. In this study we used MEFs isolated from the *B4gat1^{LacZ/M155T}* mice and measured endogenous B4GAT1 activity. As expected the *B4gat1*-deficient MEFs were hypomorphic, producing low-level residual B4GAT1 activity (<3% relative to levels in wild-type control) (**Figure 4B**). This corroborates that our B4GAT1 assay can be a valuable diagnostic tool for measuring endogenous activity in patient cells and tissues. The re-sidual B4GAT1 enzyme activity in the *B4gat1*-deficient MEFs was also reflected when the α-DG gly-cosylation status was analyzed biochemically, by immunoblotting. Although B4GAT1 endogenous activity was very low, it was sufficient to synthesize low amounts of functionally active α-DG that was capable of binding the ligand laminin (**Figure 4A**). This finding accounts for the difference between the early embryonic lethal phenotype in *B4gat1* null (*B4gat1^{LacZ/LacZ}*) mice and the slightly milder phe-notype in *B4gat1* hypomorphic (*B4gat1^{LacZ/M155T}*) mice (**Wright et al., 2012**).

It is worth noting that α-DG glycosylation is highly tissue specific as well as highly dependent on the developmental stage of the cells/tissue (**Barresi and Campbell, 2006**). To date, it is not fully under-stood what causes the tissue-specific differences in α-DG processing, which are reflected as differ-ences in its molecular weight and its ability to bind laminin (**Goddeeris et al., 2013**). LARGE, the key contributor to assembly of the terminal laminin-binding glycan, and B4GAT1 as the upstream priming enzyme are broadly expressed at the RNA level (**Figure 8—figure supplement 1**). Although both genes are similarly expressed in most tissues they are strikingly different in heart with *LARGE* expres-sion being high and *B4GAT1* being low. Based on published case reports it does not appear that *LARGE* patients (**Longman et al., 2003**; **Clarke et al., 2011**; **Meilleur et al., 2014**) are more prone to cardiac defects than other dystroglycanopathy patients. Also lower *B4GAT1* expression in the heart does not present a significant bottleneck for α-DG functional glycosylation as heart α-DG has full

ligand binding ability (*Goddeeris et al., 2013*). Therefore, the functional consequences of such unco-ordinated expression of *B4GAT1* and *LARGE* are currently unknown. It is more likely that other gene products involved in α-DG functional glycosylation can become limiting factors and that the integration of all involved players account for the tissue-specific differences of this complex and highly controlled synthesis pathway. Furthermore, whether α-DG that is not properly glycosylated possesses an as yet unidentified ligand binding activity remains unclear. LARGE has been shown to be highly tunable in the context of cancer, T-cell development and muscle regeneration (*de Bernabe et al., 2009*; *Liou et al., 2010*; *Goddeeris et al., 2013*). Repression of *LARGE* expression is responsible for the defects in DG-mediated cell adhesion that are observed in epithelium-derived cancer cells, and point to a defect of its glycosylation as a factor in cancer progression (*de Bernabe et al., 2009*). Similarly, it was demonstrated that expression of *B4GAT1* (*B3GNT1*) is absent in a IIH6-negative subpopulation (PC3-L) of an otherwise IIH6-positive human prostate cancer cell line (PC3). The loss of *B4GAT1* expression and laminin-binding by α-DG in these cells was inversely correlated with the observed malignancy and tumor progression of the prostate cancer when these cells were transplanted into SCID mice (*Bao et al., 2009*). In general, these results emphasize that proper α-DG glycosylation plays a critical role in tumor suppression.

Previous data suggested that B4GAT1 (B3GNT1) may be an integral component of various enzyme complexes, working with various glycosyltransferases that are functionally associated and involved in the same biosynthetic pathway. For example, it might work with B4GALT1 (*Lee et al., 2009*) in the synthesis of poly-N-acetyllactosamine and with LARGE (*Bao et al., 2009*) in synthesis of the α-DG laminin-binding glycan. It was also hypothesized that B4GAT1 may regulate LARGE, as B4GAT1 over-expression promoted formation of the LARGE-generated laminin-binding glycan (*Bao et al., 2009*). However, in light of data presented in this study, in particular the finding that endogenous LARGE activity is not affected in *B4gat1*-deficient cells and vice versa, it seems more likely that B4GAT1 and LARGE have independent enzyme activities (*Figure 4B/C*).

In an effort to provide additional direct evidence and further corroboration of our conclusion that a xylose present in the α-DG O-mannosyl post-phosphoryl glycan linker serves as endogenous acceptor for B4GAT1, we performed radioactive metabolic cell labeling with [3H]-xylose. The goal was to show radioactive labeling of DGFc340 expressed in *B4gat1*-deficient MEFs with [3H]-xylose, which could subsequently be released by β-xylosidase treatment. However, this type of metabolic cell labeling proved to be technically challenging since only an insignificant amount (~0.01%) of the total [3H]-xylose radioactivity was incorporated into the secreted DGFc340 fusion protein even after 4 day long-term labeling (data not shown). It is known that xylose uptake from the media into cells is poor (*Snider et al., 2002*), which in our case becomes the limiting factor and made this experimental approach not feasible. Nevertheless, we feel confident that the sum of our indirect data including B4GAT1 Xyl-β-MU acceptor specificity (*Figure 3C*), pgsl-208 DGFc340 LARGE acceptor test (*Figure 2D*), β-xylosidase B4GAT1 acceptor treatment (*Figure 6B*) and finally the in vitro synthesis of a GlcA-β1,3-Xyl-α1,3-GlcA-β1,4-Xyl-β-MU tetrasaccharide by the sequential action of B4GAT1 and LARGE (*Figure 7*) provide strong and convincing evidence that β-xylose is indeed the endogenous acceptor for B4GAT1.

In conclusion, our study has identified B4GAT1 as a xylose β1,4-glucuronyltransferase, and revealed that it contributes to the O-mannosyl post-phosphoryl glycan linker of α-DG by synthesizing a glucuronyl-xylosyl disaccharide. This is the crucial acceptor primer that is targeted by the glycosyltransferase LARGE to initiate formation of a heteropolysaccharide on α-DG that is involved in its binding to ligands. As *B4GAT1*-deficiency was linked to laminin-binding defects of α-DG in a variety of contexts, our new findings will shed light on the mechanism underlying α-DG glycosylation-deficient CMDs (*Buysse et al., 2013*; *Shaheen et al., 2013*) and tumors (*Bao et al., 2009*), and is expected to also open new therapeutic avenues for blocking the entry of pathogenic arenaviruses, including the hemorrhagic LASV into human cells (*Jae et al., 2013*).

## Materials and methods

### Subjects and samples

All tissues and patient cells were obtained and tested according to the guidelines set out by the Human Subjects Institutional Review Board of the University of Iowa; informed consent was obtained from all subjects or their legal guardians (See *Table 1*).

## Cell cultures

Cells were maintained at 37°C and 5% $CO_2$ in Dulbecco's modified Eagle's medium (DMEM) plus fetal bovine serum (FBS: 10% in the case of HEK293T cells, 20% in the case of fibroblasts from patient skin) and 2 mM glutamine, 0.5% penicillin-streptomycin (Invitrogen, Carlsbad, CA). Pro5 (wild-type) and the glycosylation-deficient CHO (Lec cells) mutant cell lines termed Lec2 and Lec8 were purchased from ATCC (*Patnaik and Stanley, 2006*). The Lec15.2 (*Maeda et al., 1998*) and ldlD (*Kingsley et al., 1986*) cell lines were kindly provided by Monty Krieger, the Lec13 (*Ohyama et al., 1998*) cells by Pamela Stanley and the pgsl-208 (*Bakker et al., 2009*) cells by Jeff Esko. These CHO cells were grown and maintained in F12 nutrition mix medium with 10% fetal bovine serum (Invitrogen) at 37°C and 5% $CO_2$. MEFs were generated from E13.5 embryos (*Table 1*) as previously described (*Xu et al., 2005*) and were maintained in DMEM supplemented with 10% FBS, 2 mM glutamine, and 0.5% penicillin-streptomycin at 37°C in 5% $CO_2$.

## [$^{32}$P] orthophosphate labeling of cells

Phosphorylation of α-DG in glycosylation-deficient fibroblasts was determined based on the incorporation of [$^{32}$P] into a secreted Fc-tagged α-DG recombinant protein, as described elsewhere (*Yoshida-Moriguchi et al., 2010*).

## Adenovirus generation and gene transfer

E1-deficient recombinant adenoviruses (Ad5CMV-DGFc340, Ad5CMV-DGFc340mut (T317A/T319A) and Ad5CMV-*LARGE*/RSVeGFP) were generated by the University of Iowa Gene Transfer Vector Core and have been described previously (*Barresi et al., 2004*). The constructs used to generate the E1-deficient recombinant adenoviruses Ad5CMV-DGFc340 and Ad5CMV-DGFc340mut (T317A/T319A) were made from pcDNA3-DGFc340 and DGFc340mut (T317A/T319A) (*Hara et al., 2011*). pcDNA3-DGFc340 vectors were digested with KpnI/XbaI, and the resulting fragments were ligated into a KpnI/XbaI-digested pacAd5-CMV-KNpA vector. Similarly, Ad5CMV-B4GAT1-V5/RSVeGFP was generated by PCR amplifying a 1.3 kb C-terminal V5-tagged open reading frame fragment corresponding to mouse *B4gat1* (*B3gnt1*, NM_175383) and cloning it into the XhoI/NotI polylinker region of pAd5CMVK-NpA. The following primers were used to amplify, by PCR, *B4gat1*-V5: 6823 forward (5′-aga**ctcgag**accATGcaaatgtcctacgccat-3′, XhoI adapter is bolded, start ATG is shown in capital letters) and 6822 reverse (5′-tat**gcggccgc**CTACGTAGAATCGAGACCGAGGAGAGGGTTAGGGATAG-GCTTACCgcatcggtggggagagttgg-3′; the NotI adapter is bolded and the V5-tag is shown in capital letters). Cultured cells were infected with viral vector for 12 hr, at an MOI of 400. We examined cultures 3–5 days after treatment. We used nucleofection as nonviral method for transferring genes into MEF cells. The Human Dermal Fibroblast Nucleofector kit was used according to an optimized protocol provided by the manufacturer (Amaxa Biosystems, Germany).

## Glycoprotein enrichment and biochemical analysis

WGA-enriched glycoproteins from frozen samples and cultured cells were processed as previously described (*Michele et al., 2002*). Immunoblotting was carried out on polyvinylidene difluoride (PVDF) membranes as previously described (*Michele et al., 2002*). Blots were developed with IR-conjugated secondary antibodies (Pierce Biotechnology, Rockford, IL) and scanned with an Odyssey infrared imaging system (LI-COR Bioscience, Lincoln, NE). Laminin overlay assays were performed as previously described (*Michele et al., 2002*).

The monoclonal antibodies to the fully glycosylated form of α-DG (IIH6) (*Ervasti and Campbell, 1991*), and also the polyclonal antibodies rabbit β-dystroglycan (AP83) (*Duclos et al., 1998*) and anti-LARGE (Rb331) (*Kanagawa et al., 2004*) were characterized previously. G6317 (core-αDG) from rabbit antiserum was raised against a keyhole limpet hemocyanin (KLH)-conjugated synthetic peptide of human dystroglycan (*Willer et al., 2012*). Mouse monoclonal anti-Myc (clone 4A6) antibodies were purchased from Millipore (Billerica, MA), mouse monoclonal anti-β-Actin (Clone AC-74) antibodies were purchased from Sigma (St. Louis, MO) and mouse monoclonal anti-V5 antibodies were purchased from Invitrogen.

## Immunofluorescence microscopy

HEK293T cells expressing Myc-tagged glycosyltranferases were fixed with 4% paraformaldehyde in PBS, and then permeabilized with 0.2% Triton X-100 in PBS for 10 min on ice. After blocking with 3% BSA in PBS, the slides were incubated with anti-c-Myc antibody (4A6, Millipore) anti-Giantin

(abcam, United Kingdom) or anti-ERp72 antibody (Calbiochem, San Diego, CA) for 18 hr at 4°C. The cells were incubated with an appropriate secondary antibody conjugated to Alexa488 or Alexa555 fluorophore after washing with PBS. 4′,6′-Diamidino-2-phenylindole dihydrochloride (DAPI, Sigma) was used for nuclear staining. Images were observed using a Zeiss Axioimager M1 fluorescence microscope (Carl Zeiss, Thornwood, NY).

## On-Cell complementation and Western Blot assay

The On-Cell complementation assay was performed as described previously (*Willer et al., 2012*). In brief, $2 \times 10^5$ cells were seeded into a 48-well dish. The next day the cells were infected with 200 MOI of Ad5CMV-*LARGE1/eGFP* in growth medium. Three days later, the cells were washed in TBS and fixed with 4% paraformaldehyde in TBS for 10 min. After blocking with 3% dry milk in TBS +0.1% Tween (TBS-T), the cells were incubated with primary antibody (glyco α-DG, IIH6) in blocking buffer overnight. To develop the On-Cell Western blots we conjugated goat anti-mouse IgM (Millipore) with IR800CW dye (LI-COR), subjected the sample to gel filtration, and isolated the labeled antibody fraction. After staining with IR800CW secondary antibody in blocking buffer, we washed the cells in TBS and scanned the 48-well plate using an Odyssey infrared imaging system (LI-COR). For cell normalization, DRAQ5 cell DNA dye (Biostatus Limited, United Kingdom) was added to the secondary antibody solution.

## Cloning of C-terminal Myc-tagged B4GAT1, TMEM5, FKTN, FKRP and LARGE

Open reading frames (ORF) were PCR amplified using the following primer sequences:

m*B4gat1* (1.3 kb), pTW324: forward 5′-aagGGATCCacc**atg**caaatgtcctacgccatccg-3′ (BamHI restriction site is shown in capital letters and start ATG is bolded) and reverse 5′-aga**gcggccgc**CTACAAGTCTTCTTCAGAAATAAGTTTTTGTTC**GCTAGC**cccgcatcggtggggagagttgggg-3′(NotI restriction site is bolded, Myc-tag sequence is underlined and NheI restriction site is shown in capital bold letters).

h*FKTN* (1.4 kb), pTW322: forward 5′-taaAGATCTacc**atg**agtagaatcaataagaacgtggttttg-3′ (BglII restriction site is shown in capital letters and start ATG is bolded) and reverse 5′-ttcGCTAGCcccatataactggataacctcatcccactc-3′ (NheI restriction site is shown in capital letters).

m*Fkrp* (1.5 kb), pTW323: forward 5′-taaGGATCCacc**atg**cggctcacccgctgctg-3′ (BamHI restriction site is shown in capital letters and start ATG is bolded) and reverse 5′-ttcGCTAGCcccaccgcctgtcaagcttaagagtgc-3′ (NheI restriction site is shown in capital letters).

m*Tmem5* (1.3 kb) pTW330: forward 5′-taaGGATCCacc**atg**cggctgacgcggacacg-3′ (BamHI restriction site is shown in capital letters and start ATG is bolded) and reverse 5′-ttcGCTAGCcccaactttattataataaaaaatgaactttc -3′(NheI restriction site is shown in capital letters).

m*Large* (2.3 kb), pTW355: forward 5′-taaAGATCTacc**atg**ctgggaatctgcagagggag-3′ (BglII restriction site is shown in capital letters and start ATG is bolded) and reverse 5′-ttcGCTAGCcccgctgttgttctcagctgtgagatatttc-3′ (NheI restriction site is shown in capital letters).

First a BamHI/NotI digested PCR fragment from m*B4gat1* was cloned into the BamHI/NotI multiple cloning site (MCS) of a pIRES-puro3-derived vector, in which the NheI site in the MCS was deleted. Subsequently all other genes were digested with either BamHI and NheI or BglII and NheI, and subcloned into a BamHI and NheI-digested m*B4gat1*-myc pIRES-puro (pTW324) construct.

## Cloning of B4GAT1-Myc Mut 1-3 mutant constructs

To generate the mouse B4GAT1-Myc Mut1-Mut3 mutant expression constructs, we used the same forward primer A (5′-aagGGATCCCaccatgcaaatgtcctacgccatccg-3′) and a reverse primer D (5′- agagcggccgcCTACAAGTCTTCTTCAGAAATAAGTTTTTGTTCGCTAGCcccgcatcggtggggagagttgggg-3′) that were used to clone m*B4GAT1*-Myc (see m*B4gat1*-Myc pTW324 cloning). Primers A and D bind at the 5′-end and 3′-end of the m*B4gat1* coding region. For each mutation we designed overlapping forward (B1-3) and reverse (C1-3) primers that included the respective mutation (shown in bold capital letters):

**mB4GAT1-Mut1** (c.1168A > G, p.N390D): B1 5′-ccaaaaggaggctgaa**G**accagcgcaataagatc-3′ and C1: 5′-gatcttattgcgctggt**C**ttcagcctccttttgg-3′.

**mB4GAT1-Mut2** (c.679/685 G > A, p.D227N/D229N): B2 5′- ggccaactacgccctggtgatt**A**atgtg**A**acatg gtgcccagcgaagggc-3′ and C2 5′- gcccttcgctgggcaccatgt**T**cacat**T**aatcaccagggcgtagtggcc-3′.

**mB4GAT1-Mut3** (c.464T > C, M155T): B3 5′- gcgctagggtcgcca**C**gcacctcgtgtgcccctc-3′ and C3 5′- gaggggcacacgaggtgc**G**tggcgaccctagcgc.

Using the m*B4gat1*-Myc (pTW324) expression construct as template, we PCR amplified 5′-fragments, using primer pairs A/B1-3 and 3′-fragments using C1-3/D, respectively. The PCR products were isolated

and used as the template DNAs in the second round of amplification with primer pair A-D. The 1.3 kb final PCR product was purified and digested with BamHI/NheI and then ligated into pTW324 digested with the same enzymes. The sequence of the insert DNA was confirmed by Sanger sequencing.

## Cloning of B4GAT1dTM

The construct expressing B4GAT1 without its transmembrane region was generated by amplifying a 1.1 kb cDNA fragment of mouse *B4gat1* (*B3gnt1*, acc.# NM_175383) from mB4GAT1-Myc expression vector pTW324, using primer pair #8629 (5'-ggtGAATTCcacggccaggaggagcagg-3') and #8630 (5'- atgACCGGTatgcatattcaagtcttcttcagaaataagtttttgttcgc-3'). EcoRI and AgeI restriction sites included in the primers are indicated in capital letters. The PCR fragment was digested with EcoRI and AgeI and subcloned to generate construct pCMV3xFLAG-TEV-B4GAT1dTM-Myc6xHIS (pTW351), which expresses a mouse B4GAT1dTM fusion protein (amino acids 37–415) tagged with a N-terminal 3xFLAG and C-terminal Myc6xHis.

## Generation of cell lines stably expressing B4GAT1dTM proteins

HEK293T cells were transfected with constructs pTW351 (B4GAT1dTM) using FuGENE 6 (Roche Applied Science, Indianapolis, IN). The construct contains an IRES-puromycin resistance cassette and stable cell lines were selected in medium containing Puromycin (1 µg/ml, InvivoGen, San Diego, CA). Expression and secretion of B4GAT1dTM protein into the culture medium was confirmed by immunoblotting with anti-Myc antibody 4A6 (Millipore). The stable cell lines obtained in this way were adapted to serum-free medium 293SFMII (Invitrogen) and cultivated in CELLine bioreactors (CL1000, Argos Technologies, Elgin, IL).

## Purification of B4GAT1dTM and LARGEdTM

B4GAT1dTM and LARGEdTM secreted into the culture medium by HEK293T cells were purified using the Talon metal-affinity resin (Clontech, Mountain View, CA) according to the manufacturer's instructions. The purity of the protein was confirmed by SDS-PAGE and Coomassie Brilliant blue (CBB) staining (*Figure 3—figure supplement 1B*). For the enzyme assay, the eluate was desalted and concentrated using an Amicon Ultra centrifugal filter unit (Millipore).

## DGFc340 in vitro LARGEdTM assay

To generate the DGFc340 and DGFc340-mut acceptor proteins we infected control and glycosylation-deficient MEFs and CHO-derived cell lines with Ad5-CMV DGFc340 adenoviral vectors at an MOI of 400. At 4 days post-infection the secreted proteins were isolated from the culture medium using Protein A-agarose beads (Santa Cruz, Dallas, TX). DGFc340 bound Protein A-agarose beads were washed three times with TBS and Protein A slurry prebound with ~25 µg DGFc340 was added to the in vitro LARGEdTM assay. Enzyme reactions (50 µl) were carried out at 37°C, with 5 mM UDP-GlcA and 5 mM UDP-Xyl, in 0.1 M MES (2-(*N*-morpholino)ethanesulfonic acid) buffer (pH 6.0) supplemented with 10 mM $MnCl_2$, 10 mM $MgCl_2$, 0.2% Triton X-100 and 5 µg purified LARGEdTM protein. The reaction was terminated by adding 5× LSB and boiling for 5 min, The samples were subsequently analyzed by immunoblotting.

## DGFc340 [$^{14}$C] radioactive sugar donor in vitro assay

DGFc340 (~25 µg) and DGFc340-mut (~25 µg) bound Protein A-agarose beads were washed with TBS and used in the in vitro LARGEdTM assay. 30 µl enzyme reactions were carried out at 37°C for 20 hr, with 0.05 µCi UDP-GlcA [GlcA-$^{14}$C] (final conc. 5.5 µM) and 0.05 µCi UDP-Xyl [Xyl-$^{14}$C] (final conc. 6.6 µM), in 0.1 M MES buffer (pH 6.0) supplemented with 10 mM $MnCl_2$, 10 mM $MgCl_2$, 0.2% Triton X-100 and 5 µg purified LARGEdTM protein. The reaction was terminated by adding 25 µl of 0.1 M EDTA. After three washes with TBS the Protein A-agarose-bound DGFc340 samples were analyzed by scintillation counting.

The reactions for B4GAT1dTM activity were carried out similarly. Again, 30 µl enzyme reactions were carried out at 37°C for 20 hr and with 0.05 µCi UDP-GlcA [GlcA-$^{14}$C] (final conc. 5.5 µM), but in this case 0.1 M MOPS (3-(*N*-morpholino)propanesulfonic acid) buffer (pH 7.0) supplemented with 10 mM $MnCl_2$, 10 mM $MgCl_2$, 0.2% Triton X-100 was used, with 0.25 µg purified B4GAT1dTM protein.

[$^{14}$C] labeled sugar nucleotides were purchased from ARC (American Radiolabeled Chemicals, St. Louis, MO).

## Glycosidase digestion

Recombinant β-glucuronidase from *E.coli* was purchased from Sigma (G8295). Each digest was performed in a 100 µl volume at 37°C for 12 hr in 50 mM NaPO$_4$, pH 7.0, 5 mM DTT, 1 mM EDTA, 0.1% Triton X-100 in the presence of 10 µg (100 units) β-glucuronidase.

Recombinant β-xylosidase from *E.coli* was purchased from Sigma (X3504). Each digest was performed in a 100 µl volume at 70°C for 60 min in 50 mM sodium acetate at pH 5.8 in the presence of 20 µg β-xylosidase.

## Analysis of enzymatic activities of B4GAT1 and LARGE

The HPLC-based enzyme assays for B4GAT1-Myc (100 µg cell lysates) and B4GAT1dTM (0.25 µg purified protein) were performed using Xyl-β-MU (0.1 mM) (Sigma) as the acceptor. The samples were incubated for 2 hr for analytical purposes and 24 hr for preparative purposes. 50 µl enzyme reactions were carried out at 37°C, with 5 mM UDP-GlcA, in 0.1 M MOPS buffer (pH 7.0) supplemented with 10 mM MnCl$_2$, 10 mM MgCl$_2$, and 0.2% Triton X-100. The reaction was terminated by adding 25 µl of 0.1 M EDTA and boiling for 5 min. The supernatant was analyzed using a *LC*18 column (4.6 × 250 mm Supelcosil LC-18 column (Supelco, Bellefonte, PA)) with Buffer A (50 mM ammonium formate pH 4.0) and Buffer B (80% acetonitrile in buffer A), using a 12% B isocratic run at 1 ml/min using Beckman Gold system (Beckman Coulter, Inc., Brea, CA). The elution of MU derivatives was monitored by fluorescence detection (325 nm for excitation, and 380 nm for emission). For the assessment of metal dependence, all ions were used at a concentration of 10 mM in 0.1 M MOPS pH 7.0. To test pH-dependent activity testing buffers ranging from pH 4.5–8.5 were used: 0.1 M sodium acetate (pH 4.5–5.5), 0.1 M MES (pH 5.5–6.5), 0.1 M MOPS (pH 6.5–7.5) and 0.1 M Tris (pH 7.5–8.5).

To assess endogenous B4GAT1 GlcA-T activity in MEFs, we solubilized the cells in TBS 1% TX-100. 100 µg total protein from crude lysates were added to each assay. 50 µl enzyme reactions were carried out for 18hr at 37°C, with 5 mM UDP-GlcA, in 0.1 M MOPS buffer (pH 7.0) supplemented with 10 mM MnCl$_2$, 10 mM MgCl$_2$, and 0.2% Triton X-100. For analysis of substrate specificity Xyl-α-MU (Sigma), Xyl-β-MU (Sigma) and Xyl-α1,3-GlcA-β-MU were added to the standard enzyme reaction at a concentration of 0.1 mM.

The HPLC-based enzymatic assay for LARGEdTM (5 µg purified protein) and endogenous LARGE was performed using GlcA-β-MU, GlcA-β1,3–Xyl-α-MU and GlcA-β1,4–Xyl-β–MU as the acceptor for Xyl-T activity and Xyl-α1,3-GlcA-β-MU for GlcA-T activity as described previously (*Inamori et al., 2012*, *2013*, *2014*). For the assessment of endogenous LARGE GlcA-T activity in MEF cells, we solubilized the cells in TBS 1% TX-100 and enriched glycoproteins from crude lysates (2 mg total protein) using WGA-agarose. N-Acetylglucosamine-eluted glycoproteins from WGA-bound glycoproteins were incubated in a volume of 50 µl for 18 hr at 37°C, with 0.1 mM MU-acceptor, 5 mM UDP-GlcA in 0.1 M MES buffer pH 6.0, 10 mM MnCl$_2$, 10 mM MgCl$_2$ and 0.2% Triton X-100. The reaction was terminated by adding 25 µl of 0.1 M EDTA and boiling for 5 min, and the supernatant was analyzed with an LC-18 column using a 12% B isocratic run.

## Analysis of B4GAT1 GlcNAc-transferase enzyme activity

The test B4GAT1dTM for GlcNAc transferase activity Gal-β1,4-GlcNAc-β-MU (0.1 mM) was used as acceptor. The 50 µl enzyme reactions were carried out as described previously (*Sasaki et al., 1997*) at 37°C, with 5 mM UDP-GlcNAc in 0.1 M cocodylate buffer (pH 7.0) supplemented with 20 mM MnCl$_2$, 5 mM ATP and 0.25 µg B4GAT1dTM enzyme. The reaction was terminated by adding 25 µl of 0.1 M EDTA and boiling for 5 min. The supernatant was analyzed using a LC18 column (4.6 × 250 mm Supelcosil LC-18 column (Supelco)) with Buffer A (50 mM ammonium formate pH 4.0) and Buffer B (80% acetonitrile in buffer A), using a 16% B isocratic run at 1 ml/min using Beckman Gold system.

## Separation and purification of the disaccharide generated by B4GAT1dTM

A large scale reaction was carried out using B4GAT1dTM purified using a metal-affinity resin as described previously for LARGEdTM (*Inamori et al., 2012*). B4GAT1dTM was added to 10 mM of UDP-GlcA and Xylose-β–MU in 50 mM MOPS buffer pH 7.0, 10 mM MgCl$_2$, 10 mM MnCl$_2$ and 0.5% TX-100 and incubated for 48 hr at 37°C with rotation. The sample was then run over a C18 column (4.6 × 250 mm Supelcosil LC-18 column (Supelco)) with Buffer A (50 mM ammonium formate pH 4.0) and Buffer B (80% acetonitrile in buffer A) using a 16% B isocratic run at 1 ml/min on a Beckman Gold

system. The elution of MU derivatives was monitored by fluorescence detection (325 nm for excitation, and 380 nm for emission). The product in the peak fractions was collected and lyophilized. The dried sample was then brought up in Milli-Q water (500 µl) and lyophilized and this procedure was repeated three times, after which the sample was brought up in Milli-Q water. The product was quantitated based on the standard curve of GlcA-β-MU. This sample was used for NMR studies.

## Separation and purification of the tetrasaccharide generated by B4GAT1dTM and LARGEdTM

The GlcA-β1,4-xylose-β-MU disaccharide (B4GAT1 product) was added to 10 mM of UDP-Xyl in 50 mM sodium acetate buffer at pH 5.5 and with 10 mM $MgCl_2$, 10 mM $MnCl_2$, 0.5% TX-100 and LARGEdTM attached to metal-affinity resin and incubated for 48 hr at 37°C with rotation. The sample was then run over a LC18 column (4.6 × 250 mm Supelcosil LC-18 column (Supelco)) with Buffer A (50 mM ammonium formate pH 4.0) and Buffer B (80% acetonitrile in buffer A) using a 16% B isocratic run at 1 ml/min on a Beckman Gold system. The elution of MU derivatives was monitored by fluorescence detection (325 nm for excitation, and 380 nm for emission). The trisaccharide peak was collected and lyophilized. The lyophilized sample was then brought up in 10 mM UDP-GlcA in 50 mM MOPS buffer pH 6.0, 10 mM $MgCl_2$, 10 mM $MnCl_2$ and 0.5% TX-100 and incubated for 48 hr at 37°C with rotation. It was again run on a C18 column with 16% B isocratic run. The product peak fraction was then collected and lyophilized. The dried sample was brought up in Milli-Q water (500 µl) and lyophilized. This procedure was repeated a total of three times. The last time the sample was brought up in Milli-Q water and the product was quantitated using a standard curve of GlcA-β-MU. This sample was used for NMR studies.

## Synthesis and purification of the disaccharide generated by B4GALT1

A large scale reaction was carried out using recombinant human B4GALT1 (purchased from R&D Systems cat# 3609-GT, Minneapolis, MN). B4GALT1 (1.5 µg) was added to 5 mM of UDP-Gal and 3 mM GlcNAc-β−MU in 50 mM Tris buffer pH 7.5, 10 mM $MgCl_2$ and 150 mM NaCl and incubated for 48 hr at 37°C with rotation. The sample was then run over a C18 column (4.6 × 250 mm Supelcosil LC-18 column (Supelco)) with Buffer A (50 mM ammonium formate pH 4.0) and Buffer B (80% acetonitrile in buffer A) using a 16% B isocratic run at 1 ml/min on a Beckman Gold system. The elution of MU derivatives was monitored by fluorescence detection (325 nm for excitation, and 380 nm for emission). Over time in the above reaction a peak was seen that ran about 1.5 min after the GlcNAc-β−MU peak at 21.5 min. This product peak was collected was and lyophilized. The dried sample was then brought up in Milli-Q water (500 µl) and lyophilized and this procedure was repeated three times, after which the sample was brought up in Milli-Q water. The product was quantitated based on the standard curve of GlcA-β-MU. This sample was used for NMR studies.

### NMR analysis

Samples were prepared for NMR by fractionation (using gel filtration and/or LC-18 chromatography) as described above, followed by the exchange of hydroxyl hydrogens by lyophilization and dissolution in 10 mM sodium phosphate buffer pH 6.5, in 100% $D_2O$. [1]H homonuclear two-dimensional DQF-COSY (*Rance et al., 1983*), TOCSY (*Braunschweiler and Ernst, 1983*), and ROESY (*Davis and Bax, 1985*) experiments, and [1]H/[13]C heteronuclear two-dimensional HMQC, HMBC, and H2BC experiments (*Nyberg et al., 2005*) were collected using a Bruker Avance II 800 MHz NMR spectrometer equipped with a sensitive cryoprobe. All NMR spectra were recorded at 25°C. The [1]H chemical shifts were referenced to 2,2-dimethyl- 2-silapentane-5-sulfonate (DSS). NMR spectra were processed using the NMRPipe software package (*Delaglio et al., 1995*) and analyzed using NMRView software (*Johnson and Blevins, 1994*).

### Acknowledgements

We thank the Gene Transfer Vector Core (UI, supported by NIH/NIDDK P30 DK 54759) for generating adenoviruses; We thank Pamela Stanley, Monty Krieger and Jeff Esko for providing us with CHO mutant cells, David Ginty for providing us with *B4gat1* (*B3gnt1*)-deficient mice, Hans v. Bokhoven for providing us with patient fibroblasts, members of the Campbell laboratory for fruitful discussions; Andrew Crimmins for technical support; Christine Blaumueller for critical reading of the manuscript. This work was supported in part by a Paul D. Wellstone Muscular Dystrophy Cooperative

Research Center Grant (1U54NS053672, KPC and TW), a MDA grant (238219, KPC and TW) and an ARRA Go Grant (1 RC2 NS069521-01, KPC and TW). KPC is an investigator of the Howard Hughes Medical Institute.

## Additional information

### Funding

| Funder | Grant reference number | Author |
|---|---|---|
| National Institutes of Health | Paul D. Wellstone Muscular Dystrophy Cooperative Research Center - 1U54NS053672 | Kevin P Campbell, Tobias Willer |
| National Institutes of Health | American Recovery and Reinvestment Act (ARRA) - 1 RC2 NS069521-01 | Kevin P Campbell, Tobias Willer |
| Muscular Dystrophy Association | 238219 | Kevin P Campbell, Tobias Willer |
| Howard Hughes Medical Institute | | Kevin P Campbell |

The funders had no role in study design, data collection and interpretation, or the decision to submit the work for publication.

### Author contributions

TW, Conception and design, Acquisition of data, Analysis and interpretation of data, Drafting or revising the article; K-I, DV, CH, GM, YH, DBVB, LY, Acquisition of data, Analysis and interpretation of data; KMW, Analysis and interpretation of data, Contributed unpublished essential data or reagents; KPC, Conception and design, Analysis and interpretation of data, Drafting or revising the article

### Ethics

Animal experimentation: Animal care, ethical usage and procedures were approved and performed in accordance with the standards set forth by the National Institutes of Health and the Animal Care Use and Review Committee at the University of Iowa (protocol #4081122). At the University of Iowa all mice are socially housed (unless single housing is required) under specific pathogen-free conditions in an AAALAC accredited animal facility. Housing conditions are as specified in the Guide for the Care and Use of Laboratory Animals (NRC). Mice are housed on Thoren brand, HEPA filtered ventilated racks, in solid bottom cages with mixed paper bedding. A standard 12/12-h light/dark cycle was used. Standard rodent chow (or special diet if required) and water is available ad libitum.

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
