## [Decision Letter]

Thank you for sending your work entitled “B4GAT1 synthesizes a glucuronyl-xylosyl acceptor required for initiation of LARGE-mediated α-dystroglycan glycosylation” for consideration at *eLife*. Your article has been favorably evaluated by Vivek Malhotra (Senior editor) and 4 reviewers, one of whom, Suzanne Pfeffer, is a member of our Board of Reviewing Editors. The Reviewing editor and the other reviewers discussed their comments before we reached this decision, and the Reviewing editor has assembled the following comments to help you prepare a revised submission.

The dystroglycanopathies are a subset of muscular dystrophies. Many result from genetic defects in the formation of a novel GlcA-Xyl glycosaminoglycan (GAG), assembled by the LARGE glycosyltransferase on a trisaccahride-Man-PO4 containing core. Knowledge of this interesting and complex receptor glycan has been driven by the identification of a number of disease genes predicted to encode novel glycosyltransferases. However, the biochemical contributions of many of the gene products, and the nature of the linker between the GAG and the Man-PO4 core, remain unknown. This manuscript 1) identifies the final sugar of the linker as a beta-GlcA, 2) shows that the beta-GlcA becomes modified by the alpha-3-XylT activity of LARGE, 3) shows that Xyl contributes to the linker, and 4) provides strong evidence that the previously identified B3GNT1 disease gene is directly responsible for addition of beta-GlcA to the underlying core, possibly forming a GlcA(beta)1-4Xyl(beta)-linkage. Overall, this is a thorough and excellently documented piece of detective work to characterize the LARGE primer on alpha-DG, in which a role for B3GNT1/B4GAT1 is systematically built up first using small molecules, then an alpha-DG surrogate and alpha-DG itself. In addition, the new activity assigned to this gene will induce reinterpretation of the basis for its interesting cellular functions in cancer. Other interesting information regarding the roles of other GT-like genes, the biochemical and cellular effects of 3 B3GNT1/B4GAT1 disease mutations, and expression of B3GNT1/B4GAT1 RNA, is also presented. The paper would, however, be strengthened if the following issues are addressed.

1) The authors put forward a stronger conclusion, that the terminal primer for LARGE is a GlcA(beta)1-4Xyl(beta)-moiety, based on detection of the corresponding activity using recombinantly prepared B3GNT1. However, there is a problem, because they do not directly demonstrate the presence of the disaccharide on native alpha-DG or DGFc340. Although they show that B3GNT1 is capable of catalyzing the transfer of GlcA from UDP-GlcA in beta-linkage to the 4 position of a beta-linked Xyl residue, this was carried out at high enzyme and substrate concentrations (ratio unspecified) that might drive alternative reactions that are not relevant at native concentrations in vivo (e.g.,: Lairson et al. (2006) Nat Chem Biol 2:724-8). While they convincingly showed that the enzyme has high selectivity toward UDP-GlcA relative to other potential donor substrates, they did not screen other possible acceptors. However, they did show that the B3GNT1 acceptor activity of DGFc340 from B3GNT1-deficient cells was inhibited by pretreatment with a commercial preparation of beta-xylosidase, but the enzyme concentration was high, release of Xyl was not verified, and alternative glycosidase activities were not excluded. Thus the conclusion that the sugar underlying GlcA is Xyl(beta) is based on converging lines of evidence that alone are rather soft. This conclusion will not be incontrovertible until the presence of the disaccharide on DGFc340 or alpha-DG is directly demonstrated, such as might be supported by mass spectrometry. The authors could quite easily label DGFc340 produced in B4GAT1 mutant MEFs with 3H-xylose, show that treatment of the captured DGFc340 substrate by beta-xylosidase releases 3H-xylose and destroys it as a substrate of recombinant B4GAT, and show that, once B4GALT1 has acted on untreated DGFc340, 3H-xylose can no longer be released by treatment with beta-xylosidase. If these data are not forthcoming (and this may well be), a qualification of their conclusions is warranted.

2) The authors also rename B3GNT1 as B4GAT1. However, they do not show that it lacks the beta-3-GlcNAc-transferase (i-antigen forming) activity originally assigned to it. While it can be argued that the latter activity was also assigned based on indirect evidence, the authors did not adequately address the potential for beta-3-GlcNAc-transferase activity because they only tested Gal(beta)-MU as an acceptor, which is not a substrate, and did not test the preferred Gal(beta)1-4GlcNAc-. If it is not too difficult, they should try to clarify this directly. The authors should explain clearly at the beginning that their data will show that the gene called B3GNT1 in the literature, in fact encodes a glucuronisyltransferase and not an N-acetylglucosaminyltransferase as previously thought.

3) While many experiments add up to strong support for the authors' conclusions re B4GAT1, their interpretation of the data in Figure 1 may or may not be the case. The data in Figure 1 show that LARGE cannot override a defect in FKRP, FKTN, B4GAT1 or TMEM5. However, the latter does not mean that all these activities act prior to LARGE in cells. The evidence that B4GAT1 acts prior to LARGE is strong and described in the manuscript. It is also clear as shown later that LARGE acting in the absence of these activities in vitro can generate laminin binding glycans. However, one or more of the remaining three activities could potentially act after LARGE and be required to modify LARGE-generated glycans such that they become capable of binding to laminin or a different binding partner when LARGE is expressed at endogenous levels in vivo. Determining whether these activities act prior to or post LARGE is beyond the scope of this manuscript. However, the authors should qualify their conclusions with respect to Figure 1 and the model in Figure 18.

---

## [Author Response]

*1) The authors put forward a stronger conclusion, that the terminal primer for LARGE is a GlcA(beta)1-4Xyl(beta)-moiety, based on detection of the corresponding activity using recombinantly prepared* B3GNT1*. However, there is a problem, because they do not directly demonstrate the presence of the disaccharide on native alpha-DG or DGFc340. Although they show that* B3GNT1 *is capable of catalyzing the transfer of GlcA from UDP-GlcA in beta-linkage to the 4 position of a beta-linked Xyl residue, this was carried out at high enzyme and substrate concentrations (ratio unspecified) that might drive alternative reactions that are not relevant at native concentrations* in vivo *(e.g.,: Lairson et al (2006) Nat Chem Biol 2:724-8). While they convincingly showed that the enzyme has high selectivity toward UDP-GlcA relative to other potential donor substrates, they did not screen other possible acceptors. However, they did show that the* B3GNT1 *acceptor activity of DGFc340 from* B3GNT1*-deficient cells was inhibited by pretreatment with a commercial preparation of beta-xylosidase, but the enzyme concentration was high, release of Xyl was not verified, and alternative glycosidase activities were not excluded. Thus the conclusion that the sugar underlying GlcA is Xyl(beta) is based on converging lines of evidence that alone are rather soft. This conclusion will not be incontrovertible until the presence of the disaccharide on DGFc340 or alpha-DG is directly demonstrated, such as might be supported by mass spectrometry. The authors could quite easily label DGFc340 produced in* B4GAT1 *mutant MEFs with 3H-xylose, show that treatment of the captured DGFc340 substrate by beta-xylosidase releases 3H-xylose and destroys it as a substrate of recombinant* B4GAT*, and show that, once* B4GALT1 *has acted on untreated DGFc340, 3H-xylose* can *no longer be released by treatment with beta-xylosidase. If these data are not forthcoming (and this may well be), a qualification of their conclusions is warranted*.

We thank the reviewers for raising an important point and suggesting some interesting additional experiments. We followed up with [3H] xylose labeling of DGFc340 in cells as suggested by the reviewers. As metabolic radioactive labeling of DGFc340 with [3H] xylose sounds very feasible on paper it proved to be technically challenging in reality. When we performed [3H] xylose metabolic cell labeling only ∼0.01% of the total radioactivity was incorporated into the secreted DGFc340 fusion protein. We validated the presence of the DGFc340 and DGFc340mut fusion protein by Western Blot, however the radioactive labeling for DGFc340 and DGFc340mut expressed in B3gnt1-deficient MEFs were indistinguishable and not significant. As stated by Snider M (Metabolic Labeling of Glycoproteins with Radioactive Sugars, Current protocols in cell biology, 2002) [3H] xylose is poorly taken up into cells creating a bottleneck that made this experimental approach not feasible.

Since the metabolic labeling of DGFc340 with [3H]-Xylose was not successful we did not include this data in the revised manuscript at this point. However, we present the data as Figure 9. If the reviewers feel this data exhibits value for the manuscript we will include the data in the final manuscript.Author response image 1.[3H] Xylose metabolic labeling of DGFc340. DGFc340 and DGFc340mut fusion proteins were enriched by Protein A-agarose from culture medium of [3H] Xylose labeled control and B4gat-deficient MEF cells. (**A**) Expression and functional glycosylation analysis of Fc fusion proteins by Coomassie brilliant blue (CBB) stain and by immunoblot. The SDS-PAGE was stained with CBB and the immunoblot was incubated with antibodies against glyco α-DG (IIH6) and anti-Fc. (**B**) The figure represents the incorporation of radiolabeled [3H] Xylose into DGFc340/DGFc340mut fusion proteins (n = 3). Error bars represent s.d. Statistical analyses were performed by two-tail Student's *t* test. ns: not significant (p > 0.05).

Alternatively the reviewers suggested validating the Xylose sugar on DGFc340 by mass spectrometry. Given the fact that only a very minor portion of the secreted DGFc340 fusion protein gets modified with the functional glycan this approach is not feasible either.

We feel that the sum of our data and especially the in vitro synthesis of a GlcA-β1,3-Xyl-α1,3-GlcA-β1,4-Xyl-β-MU -tetrasaccharide by the sequential action of B4GAT1 and LARGE (Figure 13) provide strong and convincing evidence that B4GAT1 contributes to the primer/acceptor disaccharide (GlcA-β1,4-Xyl) that can be extended by LARGE to form the heteropolymer known to bind α-DG ligands.

*2) The authors also rename* B3GNT1 *as* B4GAT1*. However, they do not show that it lacks the beta-3-GlcNAc-transferase (i-antigen forming) activity originally assigned to it. While it* can *be argued that the latter activity was also assigned based on indirect evidence, the authors did not adequately address the potential for beta-3-GlcNAc-transferase activity because they only tested Gal(beta)-MU as an acceptor, which is not a substrate, and did not test the preferred Gal(beta)1-4GlcNAc-. If it is not too difficult, they should try to clarify this directly. The authors should explain clearly at the beginning that their data will show that the gene called* B3GNT1 *in the literature, in fact encodes a glucuronisyltransferase and not an N-acetylglucosaminyltransferase as previously thought*.

We appreciate the reviewers concerns regarding B4GAT1 having beta 1,3 GlcNAc -T enzyme activity as originally described by Sasaki et al. (1997). Indeed the tested Gal-b-MU is not the preferred acceptor substrate for the reported iGnT enzyme activity. To further validate our statement that B4GAT1 has no GlcNAc-transferase activity we synthesized and purified the hypothesized preferred substrate Gal-b1,4-GlcNAc-MU in vitro, confirmed it by NMR and tested it with B4GAT1dTM as acceptor for GlcNAc transferase activity. Using the same iGnT acceptor and enzyme conditions as described in Sasaki et al. (1997) we were not able to detect any enzyme product, further confirming that B4GAT1 does not have GlcNAc transferase activity and that the enzyme originally described as iGnT / B3GNT1 had been falsely assigned.

We hope that our additional experiments included in the manuscript as Figure 15 will satisfy the reviewers and grant the renaming of the enzyme previously known as iGnT/B3GNT1 to B4GAT1.

For further clarification we also added this sentence to the Introduction: “We present experimental evidence that this enzyme B4GAT1, which was previously described in the literature as B3GNT1 (48) in fact encodes for a β1,4 glucuronyltransferase and not a β1,3 N-acetylglucosaminyltransferase as previously thought.”

*3) While many experiments add up to strong support for the authors' conclusions re* B4GAT1*, their interpretation of the data in*
Figure 1
*may or may not be the case. The data in*
Figure 1
*show that LARGE cannot override a defect in FKRP, FKTN,* B4GAT1 *or TMEM5. However, the latter does not mean that all these activities act prior to LARGE in cells. The evidence that* B4GAT1 *acts prior to LARGE is strong and described in the manuscript. It is also clear as shown later that LARGE acting in the absence of these activities* in vitro *can generate laminin binding glycans. However, one or more of the remaining three activities could potentially act after LARGE and be required to modify LARGE-generated glycans such that they become capable of binding to laminin or a different binding partner when LARGE is expressed at endogenous levels* in vivo*. Determining whether these activities act prior to or post LARGE is beyond the scope of this manuscript. However, the authors should qualify their conclusions with respect to*
Figure 1
*and the model in Figure 18*.

The reviewers bring up an interesting hypothesis that FKRP, FKTN, B4GAT1 and/or TMEM5 possibly do not act prior to LARGE glycan synthesis but rather modify the LARGE glycan itself. Patient studies have shown that all loss-of-function mutations in FKRP, FKTN, B4GAT1 and TMEM5 result in loss of aDG functional glycosylation and ligand binding. Furthermore it was demonstrated by Goddeeris et al., 2013 that the GlcA-Xyl heteropolymer synthesized by LARGE is sufficient for ligand binding. If any of the proteins in question would be involved in the post LARGE modification, mutations should not result in loss of ligand binding as the basic ligand binding LARGE glycan scaffold would still be preserved and present. Based on our current knowledge of the aDG functional glycosylation we feel that an involvement of FKRP, FKTN, B4GAT1 and/or TMEM5 in post LARGE glycan modification can be considered highly unlikely. We included a new Figure 1 and added the following statement to the figure legend: “The putative glycosyltransferases B4GAT1 (B3GNT1), FKTN, FKRP and TMEM5 are proposed to act prior to LARGE, which adds a [GlcA-Xyl] heteropolymer that is responsible for ligand binding. However based on current knowledge it cannot be completely ruled out that they are involved in the modification of the LARGE glycan repeat itself to modulate ligand binding.”